# `Web2Code`: A Large-scale Webpage-to-Code Dataset and Evaluation Framework for Multimodal LLMs

**Sukmin Yun**[*,1,4]**, Haokun Lin**[*,1]**, Rusiru Thushara**[*,1]**, Mohammad Qazim Bhat**[*,1]**,
Yongxin Wang**[*,1]**, Zutao Jiang**[1,7]**, Mingkai Deng**[2]**, Jinhong Wang**[1]**, Tianhua Tao**[1,3]**,
Junbo Li**[1]**, Haonan Li**[1]**, Preslav Nakov**[1]**, Timothy Baldwin**[1]**, Zhengzhong Liu**[1,5]**,
Eric P. Xing**[1,2,5]**, Xiaodan Liang**[1,6]**, Zhiqiang Shen**[1]
[1]MBZUAI, [2]CMU, [3]UIUC, [4]HYU ERICA, [5]Petuum, [6]SYSU, [7]Pengcheng Laboratory
https://mbzuai-llm.github.io/webpage2code/

## Abstract

Multimodal large language models (MLLMs) have shown impressive success across modalities such as image, video, and audio in a variety of understanding and generation tasks. However, current MLLMs are surprisingly poor at understanding webpage screenshots and generating their corresponding HTML code. To address this problem, we propose `Web2Code`, a benchmark consisting of a new large-scale webpage-to-code dataset for instruction tuning and an evaluation framework for the webpage understanding and HTML code translation abilities of MLLMs. For dataset construction, we leverage pretrained LLMs to enhance existing webpage-to-code datasets as well as generate a diverse pool of new webpages rendered into images. Specifically, the inputs are webpage images and instructions, while the responses are the webpage's HTML code. We further include diverse natural language QA pairs about the webpage content in the responses to enable a more comprehensive understanding of the web content. To evaluate model performance in these tasks, we develop an evaluation framework for testing MLLMs' abilities in webpage understanding and web-to-code generation. Extensive experiments show that our proposed dataset is beneficial not only to our proposed tasks but also in the general visual domain. We hope our work will contribute to the development of general MLLMs suitable for web-based content generation and task automation. Our data and code are available at https://github.com/MBZUAI-LLM/web2code.

## 1 Introduction

Multimodal large language models (MLLMs) have achieved explosive growth in the past few years. Leveraging the rich commonsense knowledge in large language models (LLMs), MLLMs are remarkably successful at processing and reasoning about various modalities such as image [2, 35], video [60, 53], and audio [40] in a broad range of tasks such as recognition [52], reasoning [59], and question-answering [39], all using language as the intermediate representation. However, existing MLLMs are surprisingly poor at understanding webpage screenshots and generating the HTML code to express their latent states. For instance, given the instruction "Parse the HTML code for this webpage", the well-known LLaVA-1.5 [33] generates generic, pale code that fails to preserve most of the original webpage's features (see Figure 1), which hampers its utility in applications such as UI prototyping, automation agents, and accessibility (e.g., noting available buttons and options given webpage screenshot).

The essential ingredients behind the progress in MLLMs are arguably large-scale instruction datasets [9, 63] and evaluation benchmarks [16, 58] – the former for aligning multimodal inputs

---

[*]Equal Contribution.

38th Conference on Neural Information Processing Systems (NeurIPS 2024) Track on Datasets and Benchmarks.

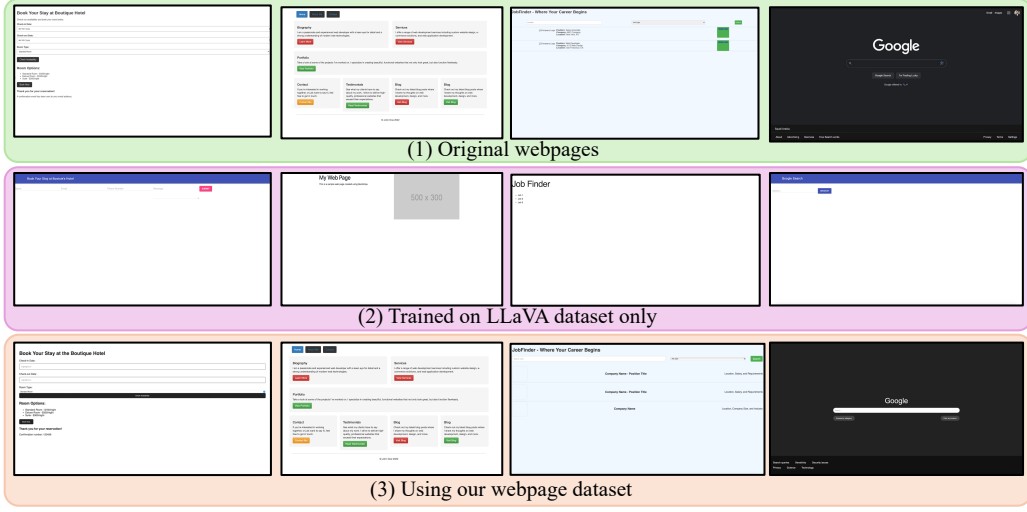

Figure 1: Our motivation for constructing the `Web2Code` dataset stems from the limitations of previous models, such as LLaVA [33], which are trained on general datasets and struggle to generate high-quality webpages, as in the second row. Our dataset aims to significantly enhance the quality of webpage generation as in third row while maintaining a strong level of general multimodal ability.

with the massive knowledge in LLMs [27, 35], and the latter for standardized comparison which facilitates model development. However, existing instruction datasets and benchmarks typically focus on general settings (e.g., visual QA and reasoning) and pay insufficient attention to webpage understanding and webpage-to-code generation, which requires a unique combination of capabilities such as optical character recognition (OCR), spatial reasoning, and long-text generation, among others. While previous work has developed datasets for these tasks [4, 22], they lack instruction information and are unsuitable for integration with general-purpose MLLMs. On the other hand, popular benchmarks [36, 25] evaluate some of the required capabilities in isolation, but not fully in combination for visual parsing and reasoning over webpages.

To fill this gap, we propose a new instruction tuning dataset and evaluation suite named `Web2Code`. `Web2Code` contains a total of 1179.7k webpage based instruction-response pairs. The responses consist of not only the HTML code, but also the structured questions and answers about the webpage, which assist a model in better understanding its information. For dataset collection, we use GPT-3.5 and GPT-4 to clean existing data (e.g. WebSRC [10]) as well as to generate completely new webpages in HTML code. To evaluate the MLLM's success at webpage understanding and HTML parsing, we propose the Webpage Understanding Benchmark (WUB) and Webpage Code Generation Benchmark (WCGB), two tasks that test the model's abilities to answer questions about a webpage and generate its HTML code, respectively. For the latter task, we find that traditional text similarity metrics are insufficient for evaluating the fidelity of the generated code, and instead propose to render the output HTML back to a webpage screenshot, and use GPT-4V [42] to evaluate the quality of the resulting webpage [61].

To demonstrate the utility of our dataset, we train LLaVA-style MLLMs with our dataset included in the instruction finetuning stage. Quantitative results show that finetuning on our dataset not only clearly improves the image-to-HTML-code translation ability of the MLLM, but also leads to improvements in the model's perception and reasoning abilities in webpage screenshot understanding as they are closely related. Furthermore, previous datasets like WebSight [22] and Pix2Code [4] present challenges: WebSight includes privacy-sensitive information, while Pix2Code contains random letters in full webpage screenshots. These issues limit MLLMs' ability to generate coherent text for webpage construction. In contrast, our dataset is more suitable for MLLM instruction fine-tuning, offering enhanced capabilities without compromising existing ones.

## 2 Related Work

**MLLM Dataset.** At present, there is a substantial amount of large-scale visual instruction data, primarily generated using GPT. SVIT [63] and LRV-Instruction [32] are both generated by GPT4

based on manual prompts to adjust the instruction data, including rich high-quality dialogue question and answer, complex reasoning question and answer, reference question and answer, and image detail description task datasets; similarly, ShareGPT4V [9], LLaVAR [62], LVIS-Instruct4V [55] use GPT-4V [42] to generate millions of high-quality image-text pairs, aiming to enhance the perception, reasoning and planning capabilities of MLLM. Commonly used image data sources include LAION [48], CC [49], SBU [46], COCO [31], VG [21], VQAv2 [18].

**MLLM.** Instruction Tuning MLLM has made great progress in recent years. The structure of MLLM usually contains a visual encoder, vision-language mapping module and LLM. LLaVA-v1.5 [33] only uses MLP as the vision-language mapping module and the successful application of instruction tuning on MLLM has inspired people. The community has explored various feasible structures, which can be divided into attention structures BLIP2 [27], InstructBLIP [12], Qwen-VL [3], ShareGPT4V [9] and non-attention structures LLaVA [33], Shikra [8] according to the vision-language mapping module. At the same time, various open source and more powerful LLMs, such as Vicuna1.5 [11], InternLM2 [5] also help MLLM achieve richer and more extensive instruction following capabilities. Qwen-VL [3], OtterHD [24], mPLUG-Owl [56], InternLM-XComposer2-4KHD [13] increase the resolution of images, while LLaVA-NeXT [34], Mini-Gemini [28], MM1 [41] split the input image into several image crops. In addition, BRAVE [19], MoVA [64], DeepSeek-VL [38], OmniFusion [17] apply supplementary vision encoders to obtain abundant visual features, e.g. DINOv2 [43], SAM [20]. Furthermore, more computer vision models are utilized for different tasks, which include image segmentation, detection and OCR, in MOAI [23], CuMo [26] and SpatialVLM [7]. Subsequently, MoE [30] was applied to MLLM to expand the scale of training data at the same computing scale.

**Code Study.** There are various code studies related to LLM. Sarker et al. [47] focuses on generating code functions using formula hints, aiming to enhance syntactic robustness and systematically testing the reliability of the syntax. From the perspective of security, Finkman et al. [15] claims that these code assistance tools may inadvertently disclose the developer's proprietary code to the code assistant service provider in the process of helping development, thus they propose a complementary method to reduce the risk of code leakage while providing effective advice to developers. In addition to code leakage, the code generated by LLM has also caused concerns in industries and other fields. To address this issue, Ye et al. [57] proposes a new zero-shot synthetic code detector based on the similarity between code and its rewritten variants. In the evaluation work on code generation, Du et al. [14] proposes a new computational efficiency benchmark Mercury and a new metric Beyond for the efficiency evaluation of code. They experimentally show that direct preference optimization can be used as a robust baseline for improving computational efficiency compared to supervised fine-tuning, which paves a promising path for future exploration of efficient code generation.

## 3 Dataset Construction

**Overview**. Our `Web2Code` instruction tuning dataset construction and instruction generation process involves four key components: **(1)** Creation of new webpage image-code pair data: We generate high-quality HTML webpage-code pairs following the CodeAlpaca prompt [6] using GPT-3.5 and convert them into instruction-following data. **(2)** Refinement of existing webpage code generation data: We transform existing datasets including WebSight [22] and Pix2Code [4] into an instruction-following data format similar to LLaVA data [33], so they can be used as instruction-following data to train MLLMs. **(3)** Creation of a new text question-answer pair data: We generate a new question-answer pair dataset utilizing our new GPT-3.5 generated data from (1) for webpage understanding. **(4)** Refinement of existing webpage understanding data: We refine the WebSRC [10] question-answer data to improve its quality using the GPT-4. Each component is elaborated in detail as follows:

**DWCG: Creation of new webpage image-code pair data for code generation.** To augment our dataset with high-quality data, we employ GPT-3.5 to generate 60K HTML pages following the guidelines and prompts in CodeAlpaca [6].[2] Using Selenium WebDriver, we then create web image screenshots from the generated HTML code. These web image-code pairs were subsequently converted into an instruction-following data format similar to the LLaVA data format [33], enabling their use in training Multimodal Large Language Models (MLLMs). The example of the instruction is shown in Figure 17. The generation of instruction is done in two stages using prompts fed to GPT-4: (a) During stage 1, the prompt shown in Figure 13 resulted in the creation of generic instructions. (b)

---

[2]To ensure both quality and diversity in the synthetic data generation, we design a prompting mechanism based on 10 detailed criteria, as detailed in Section 4.1.

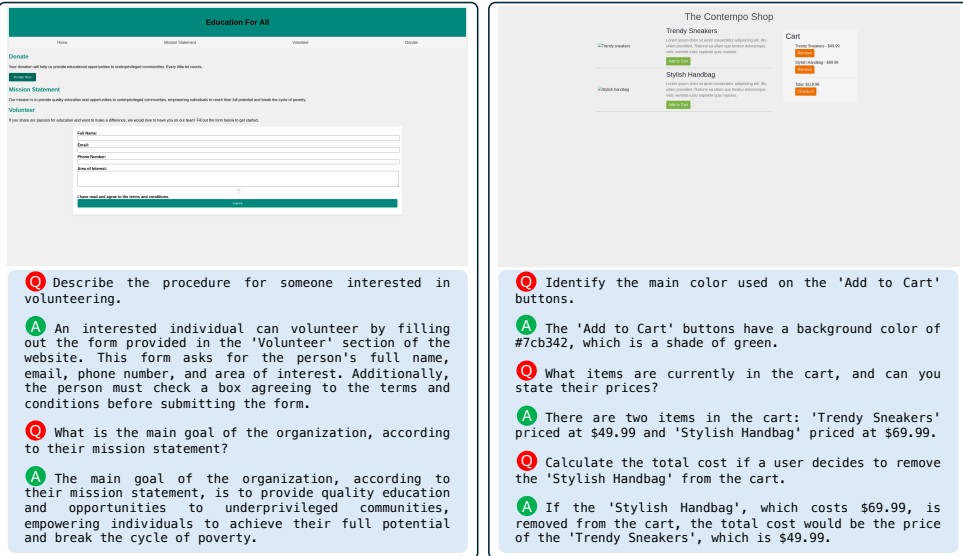

Figure 2: Qualitative example of generated question-answer pair dataset. Questions cover diverse aspects of the web page understanding.

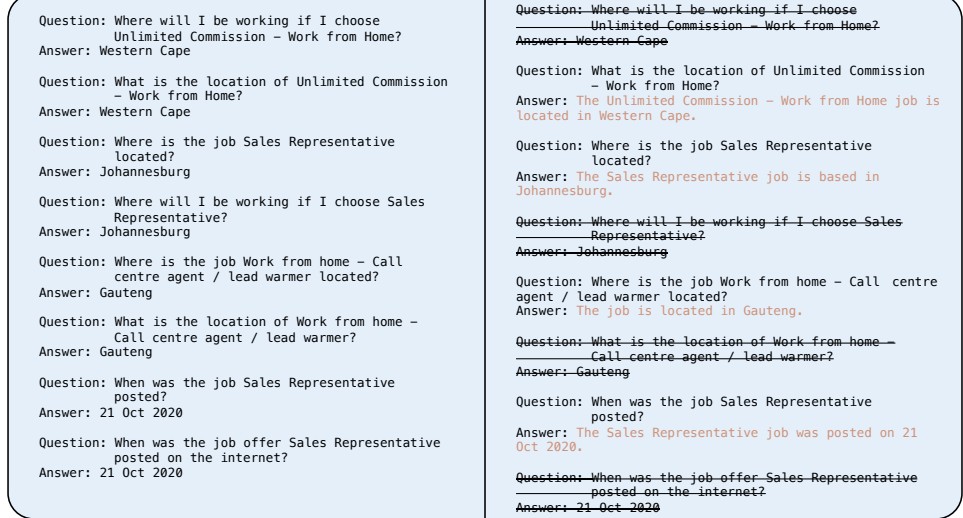

Figure 3: WebSRC data refinement for improved Quality. **Left**: Before refinement; **Right**: After refinement, the quality has been improved and duplications have been excluded.

This is followed by augmenting the instruction from (a) with the GPT generated instructions using the prompt shown in Figure 14 to include stylistic information. This allows the model to learn two styles: Modern and Bootstrap style as shown in Figure 22 and Figure 23, respectively.

**DWCG$_R$: Refinement of existing webpage code generation data.** To enhance the capability of our model in the task of HTML code generation, we leverage the Pix2code [4] and WebSight [22] datasets. To mitigate the detrimental impact on model performance from random letters in Pix2Code data, we replace these random letters with meaningful text using GPT-4, thereby refining the webpages into diverse webpages encompassing product landing pages, personal portfolios, blogs, and other categories. We then visually render each sample by taking screenshots of the browser view of each webpage. Further, we convert all these data into LLaVA instruction following data format using the same strategy as used for DWCG. We note that DWCG and WebSight webpages follow Modern style while Pix2Code follows Bootstrap style.

**DWU: Creation of a new question-answer pair data for webpage understanding.** For the purpose of fine-tuning our models in an instruction-following manner, we utilize the capabilities of GPT-4 to generate webpage code-based question-answer pairs. We generate 10 question-answer pairs using

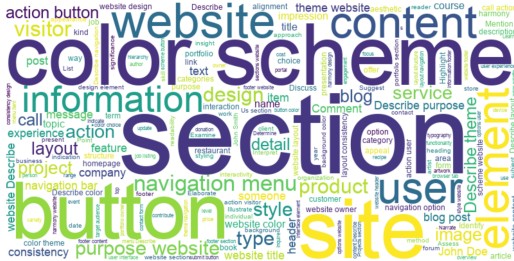
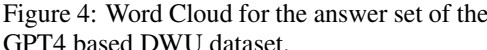

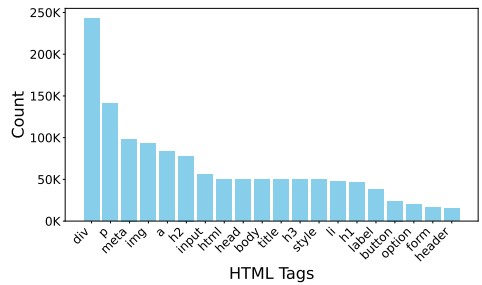

Figure 4: Word Cloud for the answer set of the GPT4 based DWU dataset.

Figure 5: Distribution of most common 20 tags in GPT-3.5 based HTML data.

| Dataset | WebSight [22] | Design2Code [50] | Pix2Code [4] | DWCG (ours) | DWCG$_R$ (ours) |
|---|---|---|---|---|---|
| **Instruction** | - | - | - | ✓ | ✓ |
| **Source** | Synthetic | Real-World | Synthetic | Synthetic | Synthetic |
| **Size** | 823K | 484 | 1.7K | 60K | 824.7K |
| **Avg Length (tokens)** | 647±216 | 31216±23902 | 658.7±98.0 | 471.8±162.3 | 652.85±157.0 |
| **Avg Tag Count** | 19±8 | 158±100 | 51.6±8.0 | 28.1±10.6 | 35.3±9.0 |
| **Avg DOM Depth** | 5±1 | 13±5 | 8.0±0.0 | 5.3±1.0 | 6.5±1.0 |
| **Avg Unique Tags** | 10±3 | 22±6 | 17.0±0.0 | 13.6±2.7 | 13.5±2.5 |

Table 1: Comparison of dataset statistics among webpage code generation datasets: WebSight, Design2Code, Pix2Code, our DWCG, and our DWCG$_R$. DWCG is a newly generated GPT-3.5-based dataset, while DWCG$_R$ is the refined dataset that utilizes WebSight and Pix2Code datasets.

GPT-4 for a subset of 24.35K webpage data, resulting in a total of 243.5K question-answer data points. This includes, a set of 230K question-answer pairs for GPT-3.5 based webpages, a set of 13.5K newly generated question answer pairs for refined Pix2Code images. These pairs are meticulously crafted to align with our image-based evaluation criteria, ensuring that each question probes specific aspects of the visual and content quality reflected in the generated web images. This strategy enhances the model's performance by integrating a nuanced understanding of the evaluation parameters into its learning process. To generate the DWU question-answer pair data, we use only the HTML code shown in Figure 12, which includes the compiled HTML image. Figure 2 shows the compiled HTML image alongside the corresponding question-answer pairs. For training, we input the image and questions together, excluding the HTML code for enhancing webpage understanding capabilities.

**DWU$_R$: Refinement of existing webpage understanding data.** To increase our instruction-following dataset with high-quality instruction-following examples for webpages, we integrate the WebSRC dataset into our training regime. Prior to inclusion, we meticulously filter the existing question-and-answer pairs from the WebSRC dataset to ensure relevance and quality. This involves duplication removal and quality optimization as shown in Figure 3. Specifically, we find that Web-SRC data contains several questions related to the same answer. To this end, we first remove those duplicates and then employed GPT-4 to assess and enhance the quality of answers. This process not only refines the dataset into 51.5K high-quality instruction data but also ensures that the model's training is influenced by high-fidelity, instructionally sound data, thereby improving its ability to follow complex web-based instructions.

### 3.1 Statistics and Analysis

Figure 4 shows the word cloud of the answer set of our question-answer dataset. The word cloud highlights the most frequently occurring terms, with "section," "color", "button", and "website" being the most prominent, indicating a strong emphasis on structural and design elements in the data. This reflects the detailed focus on the layout and visual aspects of the dataset.

Figure 5 illustrates the distribution of the most common HTML tags in our GPT-3.5 generated HTML data. The distribution shows a high frequency of essential structural tags such as <div>, <p>, <meta>, , and <a>, indicating that the generated pages include a diverse range of elements necessary for rich and varied web content. The significant presence of <h2>, <input>, <html>, <head>, and <body> tags further reinforces the completeness and structural integrity of the generated HTML documents.

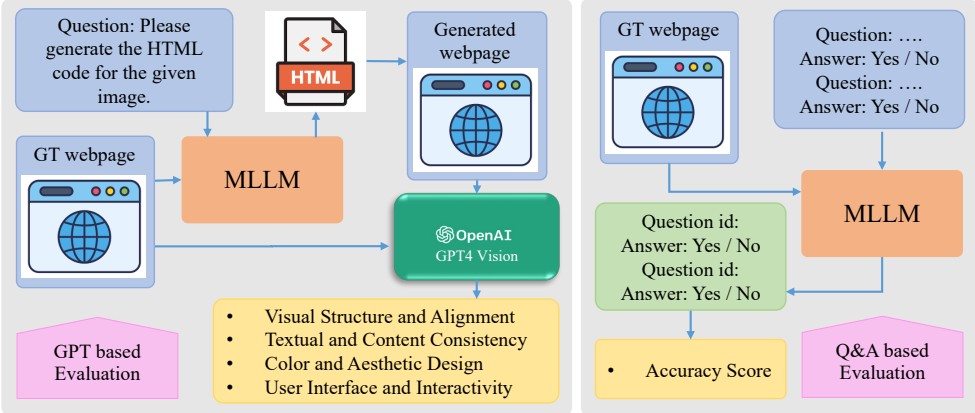

Figure 6: Evaluation benchmark for webpage generation and webpage understanding. **Left**: WCGB utilizes GPT4 Vision based online evaluation for image level comparison; **Right**: WUB employs an offline evaluation based on question-answer pairs.

To estimate the difficulty levels of our HTML-based webpage dataset, we provide several quantitative measures and compare them with recent and similar existing datasets, namely WebSight [22], Design2Code [50], and Pix2Code [4] (See Table 1).

Design2Code is primarily used for testing and has a small size of 484 examples, limiting its versatility and robustness. In contrast, our dataset, intended for both training and testing, is significantly larger (884.7K examples) and more complex, making it more suitable for developing robust models. Overall, our benchmark examples are more challenging and cover a broader spectrum of complexities compared to prior efforts such as WebSight.

## 3.2 Distribution

Our instruction-following dataset contains 1,179.7K instruction data points. This includes 884.7K website image-code pairs and 295K question-answer pairs.

| Dataset | DWU | $DWU_R$ |
|---|---|---|
| **Instruction Size** | ✓ 243.5K | ✓ 51.5K |

Table 2: Distribution of DWU and $DWU_R$ datasets. Both datasets include high-quality question-answer pairs for webpage understanding.

The 295K question-answer pairs consist of 243.5K GPT-4 based question-answer pairs (DWU Data) and 51.5K pairs from Web-SRC image-based data, as shown in Table 2. Our evaluation dataset comprises 1,198 webpage screenshot images, sourced from diverse origins, including WebSight, Pix2Code, GPT-3.5-generated data, and manual processes, to ensure a broad representation of web content. Additionally, we utilize 5,990 "yes" / "no" question-answer pairs generated from the GPT-4 Vision API for our Webpage Understanding Benchmark, as in Section 4.1.

## 4  A New Evaluation Framework for Webpage

Our proposed evaluation framework includes two schemes: (1) **W**ebpage **U**nderstanding **B**enchmark (**WUB**): An offline evaluation using "yes" / "no" questions. (2) **W**ebpage **C**ode **G**eneration **B**enchmark (**WCGB**): An online evaluation (using GPT-4 Vision) based on image similarity.

### 4.1  Evaluation Metric for HTML Code Generation

In the realm of assessing code quality, particularly in terms of final visual appeal and overall functionality, existing methods that rely on code similarity metrics fall short. These traditional approaches often lack the precision and reliability needed for nuanced evaluations of code effectiveness. To address these shortcomings, we have developed a novel approach: regenerating the webpage using the model's predicted HTML code and capturing screenshots of these generated webpages. This process, automated using the Selenium WebDriver extension in Python, shifts the focus from the less reliable code similarity assessments to a more accurate and visually oriented method. By comparing images of the generated webpages, we can more effectively evaluate the aesthetic and functional aspects of the code, offering a more comprehensive understanding of its quality.

We propose two benchmarks for assessing webpage understanding and code generation capabilities.

**WUB**: This benchmark comprises 5,990 high-quality question-answer pairs generated from GPT-4 Vision API (See prompt 16), based on 1,198 webpage screenshot images, where each answer is either "yes" or "no". These images are sourced from diverse data origins, including WebSight, Pix2Code, GPT-3.5, and manual processes, ensuring a broad representation of web content. Figure 11 shows a qualitative sample data we used for WUB. We test these pairs on various multimodal image understanding models by comparing the predicted answers to the ground truth, with the final accuracy score serving as the evaluation metric as depicted on the right side of Figure 6. Qualitative data examples in our WUB benchmark are shown in Figure 11.

**WCGB**: Utilizing the same images as the WUB, this benchmark evaluates a multimodal model tasked with generating HTML code from webpage images based on specific instructions. Unlike traditional code-level evaluations, this benchmark assesses the generated webpage's fidelity at the image level. We convert the predicted HTML codes back into images using Selenium WebDriver to allow a direct visual comparison with the ground truth images. The evaluation, depicted on the left side of Figure 6, considers 10 different aspects, which are further categorized into four evaluation matrices using the GPT-4 Vision API. This image-level evaluation provides a more accurate measure of the model's code generation capabilities, acknowledging that identical webpages can be constructed from varying codes. The prompt used for evaluation is shown in Figure 15. This framework consists of 10 distinct criteria, which we group into four categories, each encompassing specific criteria that are scored on a 0-10 scale, as follows:

1. **Visual Structure and Alignment**

   - *Layout Consistency*: Measures the arrangement of structural webpage elements like headers, footers, and sidebars.
   - *Element Alignment*: Assesses the alignment of images, buttons, and text boxes.
   - *Proportional Accuracy*: Checks for consistency in sizes and aspect ratios of visual elements.
   - *Visual Harmony*: Examines the overall balance and harmony in design.

2. **Color and Aesthetic Design**

   - *Color Scheme and Aesthetic Match*: Focuses on the similarity in color schemes, including hues and saturation.
   - *Aesthetic Resemblance*: Looks at the overall aesthetic appeal and style (modern, minimalistic, traditional, etc.).

3. **Textual and Content Consistency**

   - *Font Characteristics and Consistency*: Assesses uniformity in font type, size, style, and weight.
   - *Textual Content Match*: Evaluates the match in words and sentences.
   - *Numeric and Special Character Accuracy*: Checks for consistency in numbers, dates, and special characters.

4. **User Interface and Interactivity**

   - *User Interface Consistency*: Assesses the similarity in design language and appearance of UI elements like menus, buttons, and forms.

### 4.2 Quantitative Evaluation for HTML Code Generation of MLLMs

We have evaluated the trained models using various data configurations and backbones on our WUB and WCGB benchmarks. The performance of the models on the code generation benchmark is presented in Table 3, while the results for webpage understanding are shown in Table 4.

To be specific, our dataset components have an orthogonal contribution to the overall improvements on both the WUB and the WCGB benchmarks. Table 3 demonstrates improvements in webpage code generation quality when incrementally adding Web2Code sub-datasets; + DWCG, + DWU, + DWCG$_R$, and + DWU$_R$. For example, the results based on instruction-tuned LLaMA-3 (in the first five rows) show step-wise improvements on the WCGB benchmark from the general domain data only ($1.79 \rightarrow 6.402 \rightarrow 6.716 \rightarrow 7.806 \rightarrow 8.530$ on the overall metric). Interestingly, the instruction-tuned LLaMA-3 model trained solely on general domain data shows poor performance on WCGB,

| LLM Backbone | DWCG | DWU | DWCG$_R$ | DWU$_R$ | VSA ↑ | CAD ↑ | TCC ↑ | UII ↑ | Overall ↑ |
|---|---|---|---|---|---|---|---|---|---|
| LLaMA3-8B [1] | - | - | - | - | 1.563 | 1.777 | 1.894 | 1.911 | 1.79 |
| | ✓ | - | - | - | 5.613 | 6.575 | 6.551 | 6.870 | 6.402 |
| | ✓ | ✓ | - | - | 6.564 | 6.762 | 6.998 | 6.541 | 6.716 |
| | ✓ | ✓ | ✓ | - | 7.667 | 7.560 | 7.995 | 8.001 | 7.806 |
| | ✓ | ✓ | ✓ | ✓ | **8.522** | **8.564** | **8.421** | **8.611** | **8.530** |
| CrystalChat-7B [37] | - | - | - | - | 4.714 | 4.572 | 4.865 | 5.147 | 4.825 |
| | ✓ | ✓ | - | - | 7.900 | 8.001 | 8.204 | 8.215 | 8.080 |
| | ✓ | ✓ | ✓ | ✓ | **8.384** | **8.287** | **8.417** | **8.488** | **8.394** |
| CystalCoder-7B [37] | - | - | - | - | 3.832 | 3.678 | 3.411 | 3.992 | 3.728 |
| | ✓ | - | - | - | 7.812 | 7.899 | 8.138 | 8.112 | 7.990 |
| | ✓ | ✓ | - | - | 8.010 | **8.102** | 8.266 | 8.124 | 8.126 |
| Vicuna1.5-7B [11] | - | - | - | - | 3.042 | 3.250 | 3.333 | 3.167 | 3.198 |
| | ✓ | ✓ | ✓ | ✓ | **7.876** | **7.687** | **7.267** | **7.563** | **7.598** |

Table 3: Performance comparison of different LLM backbones under various data configurations on our Webpage Code Generation Benchmark (WCGB). "VSA" denotes Visual Structure and Alignment, "CAD" represents Color and Aesthetic Design, "TCC" represents Textual and Content Consistency, and "UII" denotes User Interface and Interactivity.

| LLM Backbone | DWCG | DWU | DWCG$_R$ | DWU$_R$ | WUB Accuracy (%) |
|---|---|---|---|---|---|
| LLaMA3-8B [1] | - | - | - | - | 65.56 |
| | ✓ | - | - | - | 60.00 |
| | ✓ | ✓ | - | - | 69.33 |
| | ✓ | ✓ | ✓ | - | 68.68 |
| | ✓ | ✓ | ✓ | ✓ | **74.84** |
| CrystalChat-7B [37] | - | - | - | - | 73.94 |
| | ✓ | ✓ | - | - | 73.48 |
| | ✓ | ✓ | ✓ | ✓ | **74.14** |
| CrystalCoder-7B [37] | - | - | - | - | 73.54 |
| | ✓ | - | - | - | 71.81 |
| | ✓ | ✓ | - | - | **73.74** |
| Vicuna1.5-7B [11] | - | - | - | - | 71.12 |
| | ✓ | ✓ | ✓ | ✓ | **71.23** |

Table 4: Accuracy of webpage understanding under various data configurations and LLM backbones. All models are instruction-tuned and evaluated on our WUB benchmark. We note that the general domain data (i.e., LLaVA) is included in all data configuration as default.

while achieves comparable performance on WUB when compared to models trained with additional webpage datasets. (see Table 4). Similar trends are also found in other LLM backbones while adding the proposed dataset shows significant improvements. Table 4 further demonstrates the effectiveness of the proposed dataset on the webpage comprehension capabilities. For example, the DWCG dataset improves code generation capabilities, though it requires more webpage understanding. However, the DWU dataset not only recovers but also enhances both WUB and WCGB performances. Moreover, the refined dataset DWCG$_R$ primarily boosts WCGB, while DWU$_R$ shows improvements across all metrics. Overall, we found the proposed dataset can enhance both webpage understanding capability and webpage code generation abilities under various LLM backbones, and LLaMA3-8B archives the best performance among all on both webpage code generation and webpage understanding.

### 4.3 Visualizations for Qualitative Evaluation

As shown in Figure 7, we compare the results between the original image which is the real-world webpage sample, the rendered image generated by using LLM backbones of Vicuna1.5-7B and CrystalChat-7B, respectively. CrystalChat-7B is a code-enhanced LLM and our visualization demonstrates that it achieves the better quality of generation than Vicuna1.5-7B even though the performance is slightly worse on the general multimodal domain, as presented in Table 6. Moreover, as in Figure 8, our rendered webpage from the model trained on our web dataset closely resembles the original image, indicating the positive impact of the `web2code` dataset. We further visualize our generation in Figure 9 when the input is a hand-drawn webpage to examine the adaptation ability of our model.

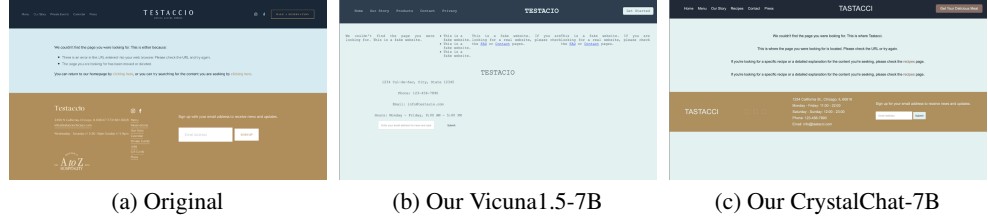

(a) Original       (b) Our Vicuna1.5-7B       (c) Our CrystalChat-7B

Figure 7: Visualization comparison using different backbones. Using the code-enhanced LLM backbone CrystalChat-7B achieves better quality of generation than Vicuna1.5-7B.

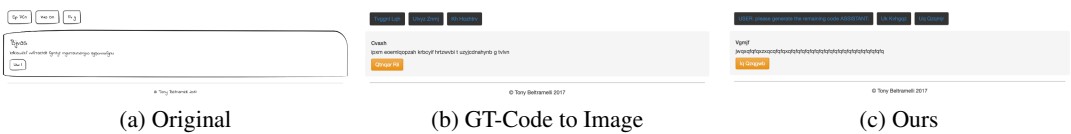

(a) Original       (b) GT-Code to Image       (c) Ours

Figure 8: Visualization comparison between ground-truth code generated image and our result. The style and layout of the generated webpage image are similar to the ground-truth image.

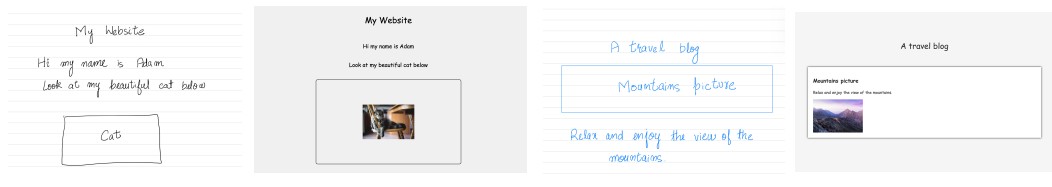

(a) Hand drawn webpage and our generation       (b) Hand drawn webpage and our generation

Figure 9: Visualization of our CrystalChat-7B generation when the input is a hand-drawn webpage.

# 5    General Evaluation of MLLMs Using `Web2Code`

**Setup and Overview.** Our model training framework mainly follows the design of LLaVA-1.5 [33] where we leverage the capabilities of both a pre-trained visual encoder, an LLM and a projector to connect visual features into the word embedding space. The model consists of (1) a pre-trained CLIP ViT-L/14 [44] visual encoder with a resolution of 336×336 and a patch size of 14, which has good feature representation already aligned with the text embedding space. (2) As for the LLM backbones, we leverage CrystalChat [37] as the base model and compare it with other latest LLM backbones like Vicuna1.5 [11], LLaMA2 [54], LLaMA3 [1] and CrystalCoder [37].[3] Training details and hyperparameters are presented in the Appendix A.

**General Evaluation Metrics for MLLMs.** MME [16] serves as an extensive evaluative benchmark, aiming to assess the perceptual and cognitive capability of MLLMs within 14 sub-tasks. Additionally, we also evaluate the performance of our models on text-oriented visual question-answering tasks employing a diverse set of benchmark datasets including ScienceQA [39] and TextVQA [51]. Furthermore, We assess our models' ability toward anti-hallucination through POPE [29].

**Effects of `Web2Code` on General Domain.** Here, we first perform instruction tuning using `Web2Code` on various LLM backbones and then we evaluate those MLLMs on the general domain of visual language understanding. Throughout extensive experiments under various data configurations, we observed that the proposed dataset `Web2Code` can be incorporated with the conventional visual language instruction tuning dataset of LLaVA [33] without harming performances on the general domain. Table 5 summarizes the results.[4] Specifically, both proposed Web Understanding data (DWU

---

[3]CrystalCoder [37] and CrystalChat [37] are an open-source code LLM pre-trained and instruction-tuned models, respectively. They are trained on publicly available language and code datasets.

[4]We observe that the conventional visual language domain data (i.e., LLaVA) is a crucial component for visual language understanding, i.e., instruction-tuned MLLMs without the general domain data are weak.

| LLM Backbone | DWU | DWCG | DWU$_R$ | DWCG$_R$ | MME-P | MME-C | POPE | SciQA | TextVQA |
|---|---|---|---|---|---|---|---|---|---|
| | - | - | - | - | 1456.53 | **308.21** | 86.86 | 67.77 | 57.84 |
| **CrystalChat-7B** [37] | ✓ | - | - | - | 1438.51 | 292.14 | **87.10** | 68.27 | **58.15** |
| | ✓ | ✓ | - | - | **1478.82** | 297.14 | 86.13 | 67.92 | 57.41 |
| | ✓ | ✓ | ✓ | ✓ | 1449.54 | 279.64 | 86.53 | **68.32** | 57.86 |

Table 5: Component analysis on CrystalChat-7B backbone under various data configurations. We note that the general domain data (i.e., LLaVA) is included in all data configuration as default.

or DWU$_R$) and Web Code Generation data (DWCG or DWCG$_R$) do not hurt or even can be beneficial to the visual language understanding. For example, we observed that adding DWU to CrystalChat achieves comparable or even better performances on POPE (86.86→87.10), SciQA (67.77→68.27), and TextVQA (57.84→58.15). Somewhat surprisingly, we further found that adding DWCG can even improve visual language understanding. For example, the second and third rows of CrystalChat show +40.31 and +5.00 points higher improvements in MME-P and MME-C benchmarks, respectively. Moreover, adding refined data DWU$_R$ and DWCG$_R$ are still effective in the visual language domain, by achieving comparable (or even better) performances on overall benchmarks. For example, the last row indicates that adding DWU$_R$ and DWCG$_R$ preserves comparable performances on overall benchmarks and even achieves +0.4 higher points on the SciQA benchmark.

## 6    Conclusion

We have presented `Web2Code`, a benchmark that consists of a high-quality, large-scale webpage-to-code instruction tuning dataset containing 1.18M entries and an evaluation suite for the webpage understanding and webpage-to-HTML translation abilities of MLLMs. To mitigate potential data biases, we have guided the synthetic data generation process toward a balanced output and incorporated diverse existing datasets to further address bias concerns. Through extensive experiments, we have demonstrated that our proposed dataset is clearly effective at enhancing these abilities of MLLMs as well as general visual proficiency, while existing datasets lead to inferior performance. We hope our work will attract the community's attention and facilitate progress toward foundation models serving as virtual assistants for content generation and task automation.

**Limitations and Ethics Statement.** The `Web2Code` project provides a comprehensive dataset and evaluation framework for fine-grained multimodal large language models. This can significantly enhance the capabilities of LLMs in understanding and generating web code from instructions, leading to advancements in web development automation, and improved coding assistance tools and platforms. By enabling more accurate and context-aware code generation, it can boost productivity for developers and make coding more accessible to beginners. However, the primary limitations of the `Web2Code` include the potential for a biased dataset that may not cover all possible HTML coding scenarios, potentially leading to gaps in model performance, and some webpages that include humans may be privacy sensitive, Ensuring high-quality annotations and comprehensive coverage of all possible HTML and code structures is challenging. Also, handling complex, real-world HTML and code scenarios might still be beyond the current capabilities of the models trained on this dataset. Moreover, the proposed evaluation framework may not capture all aspects of the code generation quality, such as code efficiency, readability, or adherence to best practices.

## Acknowledgements

We are thankful to Veselin Stoyanov for discussions on multimodal large language models. We also thank the support provided by the MBZUAI IT/corporate service teams (Ian Mathews, John Murphy, Padma Pavani, Tapas Sen, Walid Omari) and CIAI engineering team (Guowei He, Yun Xu, Yue Peng) for organizing High Performance Computing resources and services. Z.S. and E.X. would like to thank the MBZUAI-WIS Joint Program for AI Research. Z.S. and S.Y. also would like to thank the Google Research award grant and the Gemma Academic Program grant for their support. S.Y. was supported by the research fund of Hanyang University (HY-2024-2693). S.Y. was also partly supported by Institute of Information & Communications Technology Planning & Evaluation (IITP) grant funded by the Korean government (MSIT) (No.RS-2022-00155885, Artificial Intelligence Convergence Innovation Human Resources Development (Hanyang University ERICA)).

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

# Appendix

## A  Training Details and Hyperparameters

We follow the instruction-tuning protocol of LLaVA-1.5 [33]. In the pretraining step, we employ the caption data to optimize the projector, while keeping the vision encoder and LLM frozen. Meanwhile, we optimize the projector and LLM in the instruction tuning step. During the pretraining phase, we utilize a batch size of 256, while for the instruction tuning phase, we employ a batch size of 128. The learning rate is set at $1e^{-3}$ during pretraining and adjusted to $2e^{-5}$ for instruction tuning, with both phases incorporating a cosine decay schedule. We also apply a learning rate warmup with a decay factor of 0.03, and no weight decay is used. Both pretraining and instruction tuning are conducted for one epoch each, consistently using the AdamW optimizer.

## B  More Effects of `Web2Code` on General Domain

Here, we first perform instruction tuning using `Web2Code` on various LLM backbones and then we evaluate those MLLMs on the general domain of visual language understanding.

**Comparison on different LLM backbones.** We compare the general domain abilities of various LLM backbones under the same data configuration of LLaVA + DWU + DWCG; Table 6 summarizes the results of instruction-tuned MLLMs. Specifically, we found that instruction-tuned CrystalChat-7B, Vicuna1.5-7B, and LLaMA2-7B show superior performances in the general domain compared to CrystalCoder and CodeLlama. For example, CrystalChat shows +132.89 points higher than CodeLlama in MME-P (i.e. perception domain). Somewhat surprisingly, instruction-tuned CrystalChat showed the strongest performance on TextVQA, which requires visual reasoning based on text in images.

| LLM Backbone | MME-P | MME-C | POPE | SciQA | TextVQA |
|---|---|---|---|---|---|
| CodeLlama-7B [45] | 1345.93 | 258.92 | 85.28 | 61.87 | 55.23 |
| CrystalCoder-7B [37] | 1351.22 | 274.64 | 86.05 | 61.63 | 50.11 |
| CrystalChat-7B [37] | 1478.82 | 297.14 | 86.14 | 67.92 | **57.41** |
| Vicuna1.5-7B [11] | **1488.26** | 268.21 | **87.05** | **69.31** | 56.40 |
| LLaMA2-7B [54] | 1448.58 | **304.29** | 86.87 | 67.87 | 55.62 |

Table 6: Comparison of different LLM backbones on visual language understanding benchmarks. All models are instruction-tuned on the general domain data (i.e., LLaVA) with DWU and DWCG.

## C  Data collection of `Web2Code`

**Diversity.** The dataset `Web2Code` was curated from a wide variety of sources, including different types of websites and design patterns, to ensure broad coverage of real-life scenarios, Figures 4 and 5 also illustrate the diversity of our data. In Table 7 and Figure 10, we have calculated the distribution percentages of different types of web pages in our dataset to illustrate its comprehensiveness further.

| Corporate | Portfolio | Contact | Personal | Advertisement | Landing |
|---|---|---|---|---|---|
| 10.83 | 4.77 | 9.17 | 0.52 | 20.68 | 10.23 |
| Blog | E-commerce | Educational | Non-profit | Login | Entertainment |
| 8.21 | 10.77 | 2.78 | 2.54 | 1.02 | 1.74 |
| News | Forum | Event | Documentation | Marketing | Unknown |
| 0.69 | 1.89 | 0.81 | 0.15 | 0.05 | 13.14 |

Table 7: Data distribution percentages (%) of different types by category.

**Privacy.** We have implemented several measures to ensure that our data collection and processing adhere to legal and ethical standards for safeguarding privacy. Below, we provide a detailed overview of these privacy protection measures:

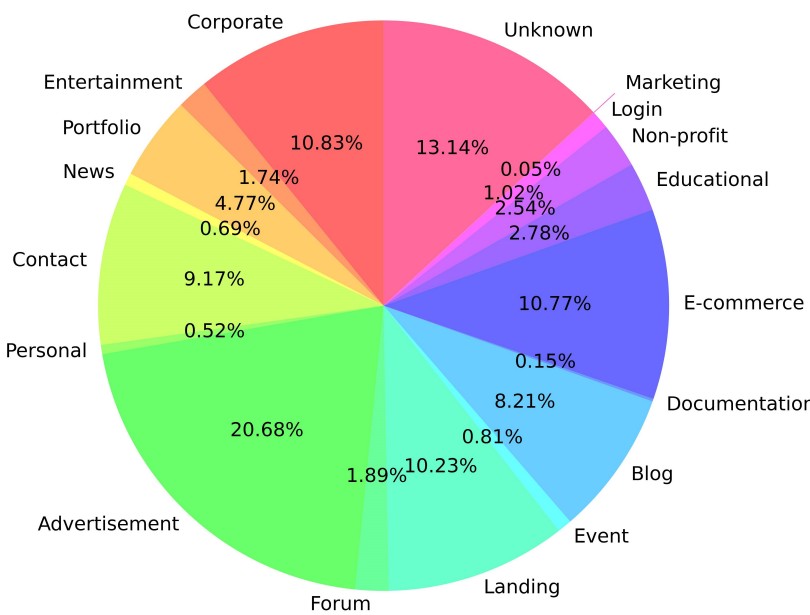

Figure 10: Data distribution percentages (%) of different types by category.

1. **Personal data removal.** During the data collection process, any personal or sensitive information (e.g., names, email addresses, phone numbers) that could be linked to individuals was systematically removed. This anonymization process ensures that the dataset does not contain any information that could directly identify users.

2. **Scrubbing of sensitive fields.** Fields known to potentially hold sensitive information, such as form inputs or user comments, were either excluded from the dataset or anonymized to prevent privacy breaches.

3. **Publicly available data.** The dataset was curated and refined primarily from publicly available datasets where no authentication is required. This reduces the risk of capturing sensitive, private content that is not intended for public consumption.

4. **Exclusion of personally identifiable information.** No personally identifiable information was included in the dataset. Any data fields that could inadvertently contain personally identifiable information were carefully reviewed and excluded or anonymized.

To further assess privacy concerns, we conducted a thorough examination by randomly sampling 1,000 data points from our dataset and performing a human evaluation to check for privacy-sensitive information, such as phone numbers, email addresses, credit card numbers, and personal images. We find that only 2.7% (16 email addresses, 11 phone numbers, and no credit card numbers) of the data contained such issues, indicating the effectiveness of our privacy protection measures.

| LLM Backbone | DWCG | DWU | DWCG$_R$ | DWU$_R$ | Block-Match ↑ | Text ↑ | Position ↑ | Color ↑ | CLIP ↑ |
|---|---|---|---|---|---|---|---|---|---|
| | - | - | - | - | 5.3 | 11.6 | 7.4 | 26.0 | 73.2 |
| | ✓ | - | - | - | 76.3 | 92.0 | 74.8 | 63.7 | 84.2 |
| LLaMA3-8B [1] | ✓ | ✓ | - | - | 75.4 | 91.3 | **77.3** | 64.9 | 85.6 |
| | ✓ | ✓ | ✓ | - | **77.1** | 94.8 | 76.7 | 64.4 | 86.3 |
| | ✓ | ✓ | ✓ | ✓ | **77.1** | **96.9** | 77.0 | **66.2** | **86.7** |

Table 8: LLaMA3-8B results on low-level element matching (Design2Code [50]) metrics and CLIP [44] score.

## D  More Analysis on Other Metrics for HTML Code Generation of MLLMs

In this section, we incorporate low-level element matching metrics from Design2Code [50] and CLIP-based assessments [44] for visual-text alignment, to provide a more comprehensive evaluation. As shown in Table 8, our proposed DWCG method shows significant improvements across various metrics, with DWU dataset enhancing performance in block-match, position, and color attributes. Models trained with $DWCG_R$ and $DWU_R$ further demonstrate improvements across all aspects.

## E  More Comparisons with Existing Datasets

Here, we have provided more comparisons with additional baselines of the WebSight datasets [22]. In Table 9, the single HTML code generation dataset (i.e., WebSight v0.1) significantly drops the performances of WUB compared to the original LLaVA dataset (without any additional code-related data). We also observe that even a single DWCG dataset (our 60K generated webpage code generation data) outperforms the WebSight v0.1 dataset of 823K. Table 10 also shows the effectiveness of the proposed dataset DWCG and DWU. Similar to the trend shown in Table 9, our 60K generated data, DWCG, significantly outperforms ∼14 times larger dataset baseline, WebSight0.1, across all low-level element matching metrics (Design2Code [50]) and CLIP [44] score.

| DWCG | DWU | WebSight v0.1 | WebSight v0.2 | VSA ↑ | CAD ↑ | TCC ↑ | UII ↑ | WCGB Overall ↑ | WUB Acc. |
|---|---|---|---|---|---|---|---|---|---|
| - | - | - | - | 3.832 | 3.678 | 3.411 | 3.992 | 3.728 | 73.54 |
| ✓ | - | - | - | 7.812 | 7.899 | 8.138 | 8.112 | 7.990 | 71.81 |
| ✓ | ✓ | - | - | **8.010** | **8.102** | **8.266** | **8.124** | **8.126** | **73.74** |
| - | - | ✓ | - | 7.524 | 7.600 | 7.818 | 7.862 | 7.701 | 59.38 |
| - | - | - | ✓ | 5.469 | 5.830 | 6.113 | 5.911 | 5.831 | 56.06 |

Table 9: WebSight baseline results compared to the DWCG and DWU datasets on the WCGB and WUB benchmarks. All models are trained on the CrystalChat-7B [37] backbone.

| DWCG | DWU | WebSight v0.1 | WebSight v0.2 | Block-Match ↑ | Text ↑ | Position ↑ | Color ↑ | CLIP ↑ |
|---|---|---|---|---|---|---|---|---|
| - | - | - | - | 8.6 | 13.2 | 6.5 | 17.4 | 70.3 |
| ✓ | - | - | - | **76.6** | 90.4 | **78.2** | 63.1 | **85.2** |
| ✓ | ✓ | - | - | 75.4 | **91.7** | 77.4 | **64.0** | 83.9 |
| - | - | ✓ | - | 62.4 | 77.7 | 63.9 | 60.7 | 79.2 |
| - | - | - | ✓ | 61.8 | 78.2 | 62.9 | 62.1 | 78.7 |

Table 10: WebSight baseline resultson low-level element matching (Design2Code [50]) metrics and CLIP [44] score. All models are trained on the CrystalChat-7B [37] backbone.

Furthermore, Tables 11, 12, and 13, summarize comparisons between the Webstight dataset and the proposed dataset, Web2Code, on various evaluation metrics, including WCGB, WUB, Design2Code, CLIP, and general domain benchmarks. Overall, our Web2Code dataset significantly outperforms the WebSight baselines across all metrics. Such degradation of existing code generation datasets in the general domain motivates us to explore ways to refine them and simultaneously collect (or generate) both code generation data and webpage understanding data to maintain overall performance.

| Web2Code | WebSight v0.1 | WebSight v0.2 | VSA ↑ | CAD ↑ | TCC ↑ | UII ↑ | WCGB Overall ↑ | WUB Acc. |
|---|---|---|---|---|---|---|---|---|
| - | - | - | 1.563 | 1.777 | 1.894 | 1.911 | 1.790 | 65.56 |
| ✓ | - | - | **8.522** | **8.564** | **8.421** | **8.611** | **8.530** | **74.84** |
| - | ✓ | - | 4.236 | 4.113 | 3.981 | 4.168 | 4.125 | 70.13 |
| - | - | ✓ | 4.611 | 5.238 | 5.923 | 4.689 | 5.115 | 57.33 |

Table 11: Comparisons between the WebSight datasets and the proposed Web2Code dataset on the WCGB and WUB benchmarks. All models are trained on the LLaMA3-8B [1] backbone.

## F  Qualitative Data Examples in WUB Benchmark

The qualitative data examples in our WUB benchmark are shown in Figure 11. It covers different aspects of webpage understanding based on "yes" / "no" question-answer pairs.

| Web2Code | WebSight v0.1 | WebSight v0.2 | Block-Match ↑ | Text ↑ | Position ↑ | Color ↑ | CLIP ↑ |
|---|---|---|---|---|---|---|---|
| - | - | - | 5.3 | 11.6 | 7.4 | 26.0 | 73.2 |
| ✓ | - | - | **77.1** | **96.9** | **77.0** | **66.2** | **86.7** |
| - | ✓ | - | 60.1 | 78.5 | 60.9 | 60.3 | 79.5 |
| - | - | ✓ | 57.3 | 73.9 | 59.1 | 60.8 | 76.1 |

Table 12: Comparisons between the WebSight datasets and the proposed `Web2Code` dataset on low-level element matching (Design2Code [50]) metrics and CLIP [44] score. All models are trained on the LLaMA3-8B [1] backbone.

| Web2Code | WebSight v0.1 | WebSight v0.2 | MME-P | MME-C | POPE | SciQA | TextVQA |
|---|---|---|---|---|---|---|---|
| ✓ | - | - | **1464.73** | 265.14 | **86.34** | **72.08** | **58.76** |
| - | ✓ | - | 1349.33 | 235.47 | 85.73 | 61.42 | 23.38 |
| - | - | ✓ | 1149.68 | **270.36** | 83.29 | 56.19 | 20.51 |

Table 13: Comparisons between the WebSight datasets and the proposed `Web2Code` dataset on general domain benchmarks. All models are trained on the LLaMA3-8B [1] backbone.

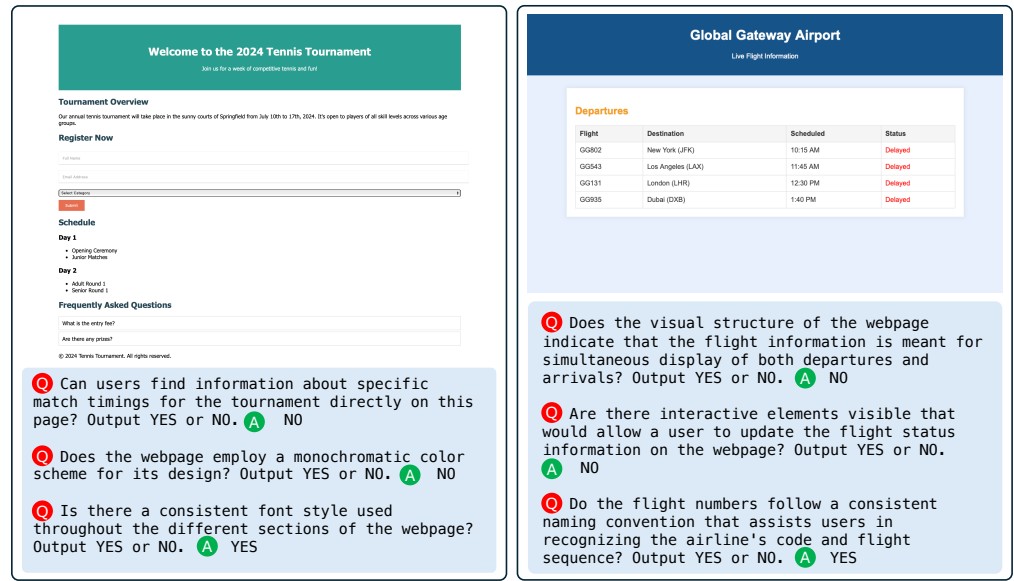

Figure 11: Qualitative data examples in our WUB benchmark. It covers different aspects of webpage understanding based on "yes" / "no" question-answer pairs.

# G  Prompt Templates

## G.1  Prompt Used to Question and Answer Generation for DWU Data

**System:** You are an advanced AI model who can identify html code and interpret the compiled webpage.

**User:** You are asked to come up with a set of 10 diverse website understanding task instructions with the corresponding source codes. These task instructions will be given to a GPT model and we will evaluate the GPT model for completing the instructions.

You should output 10 questions and the answers for them, for each html code.
Your output should be formatted as follows. Don't include any other text, connective phases other than the formatted text.
output a comma separated list of 10 instruction pairs in following format
[{"Q": "<question>" , "A": "<answer>"}, {"Q": "<question>" , "A": "<answer>"}]

Here are the requirements:
1. Try not to repeat the verb for each instruction to maximize diversity.
2. The languages used for the instruction should be diverse. For example, you should combine questions with imperative instructions.
3. The instructions should be in English.
4. The instructions should be at least 1 to 2 sentences long. Either an imperative sentence or a question is permitted.
5. Instructions should be clear and precise.
6. Instructions can either be simple queries like 'what text is displayed on the button?', to more complex tasks requiring analysis, such as 'what would be the total cost of ordering items A and B from the menu?'.
7. Avoid direct html code related instructions. i.e. Don't include questions related to html tags, font sizes, and etc. Instead include real webpage image related instructions that a human can ask after looking into the webpage. For example, What is the main event that this website is allowing users to register for?, State the variety of dishes featured in the desserts section of the menu.
8. The Answer should be an appropriate response to the instruction and the input.
9. If the answer isn't immediately clear from the content, the output should detail the thought process. For instance, if the task is to determine the age of someone born in 1983, the response should explain: 'Considering the current year is 2024, the age would be 2023 minus 1984, which equals 40.'
10. Try to make questions covering the following aspects i.e. Visual Structure and Alignment: Layout Consistency: Evaluates the match in the placement of headers, footers, and sidebars. Element Alignment: Assesses alignment of images, buttons, and text boxes. Proportional Accuracy: Checks for consistency in sizes and aspect ratios of visual elements. Visual Harmony: Examines the overall balance and harmony in design. Color and Aesthetic Design: Color Scheme and Aesthetic Match: Focuses on the similarity in color schemes, including hues and saturation. Aesthetic Resemblance: Looks at the overall aesthetic appeal and style (modern, minimalistic, traditional, etc.). Textual and Content Consistency: Font Characteristics and Consistency: Assesses uniformity in font type, size, style, and weight. Textual Content Match: Evaluates the match in words and sentences. Numeric and Special Character Accuracy: Checks for consistency in numbers, dates, and special characters. User Interface and Interactivity: User Interface Consistency: Assesses the similarity in design language and appearance of UI elements like menus, buttons, and forms.
11. Answers should be more informative and descriptive. Don't output single word or numbers for the outputs, make them descriptive with one or two sentences.
12. Don't shrink the output. Need everything expanded. Avoid outputs like (...)

Here's the html code.

```
<!DOCTYPE html>
<html>
<head>
<title>Charity Walk Registration</title>
</head>
<body>
<h1>Welcome to the Charity Walk Registration Site</h1> ...
```

Figure 12: Prompt used to generate Question Answer pairs using GPT4 for DWU data.

## G.2 Prompts Used For Instruction generation of DWCG and DWCG$_R$

```
User:
Here are some instructions that are being fed to a generative AI model as instruction tuning during the
training of a Vision language model.

"In the provided webpage screenshot, generate HTML to replicate the layout and styling of the webpage",

"Given the web application interface shown, write HTML code to implement the interactive features visible in
the image",

"Analyze the webpage structure in the screenshot and provide HTML and CSS code to design a responsive webpage
with similar layout and components",

"Examine the provided webpage screenshot and generate HTML code to implement the responsive layout",

"Given the webpage screenshot, provide the HTML code that represents its structure",

"Create HTML code for the webpage depicted in the image provided",
"Craft HTML code to replicate the visual design and structure of the webpage captured in the provided
screenshot",

"Create HTML and CSS code to imitate the appearance and layout of the web interface shown in the provided
image",

"Generate HTML and CSS code snippets to mirror the layout and styling of the website visible in the given
screenshot"

Note:
1.Try not to repeat the verb for each instruction to maximize diversity.
2.The languages used for the instruction should be diverse. For example, you should combine questions with
imperative instructions.
3.The instructions should be in English.
4.The instructions should be at least 1 to 2 sentences long. Either an imperative sentence or a question is
permitted.
5.Instructions should be clear and precise.
6.Please provide 150 examples.
```

Figure 13: Prompt used to generate instructions for DWCG using GPT4, feeding input as Seed instructions and output as GPT generated instructions shown in Figure 17.

```
User:
Here are some instructions that are being fed to a generative AI model as instruction tuning during the
training of a Vision language model.

"Please provide the code in material design style.",

"Share the code adhering to the principles of material design.",

"Ensure the code follows the guidelines of material design aesthetics.",

"Provide the code in a style consistent with material design principles.",

"Code should reflect the design language of material design.",

"Maintain a coding style aligned with material design aesthetics.",

"Present the code in accordance with material design styling."

Note:
1.Try not to repeat the verb for each instruction to maximize diversity.
2.The languages used for the instruction should be diverse. For example, you should combine questions with
imperative instructions.
3.The instructions should be in English.
4.The instruction should be clear.
5.Please provide 40 examples.
```

Figure 14: Prompt used to generate webpage style instruction for DWCG using GPT4, feeding input as Seed instructions and output as GPT generated webpage style instructions shown in Figure 17.

## G.3 Prompt Used For GPT4-Vision Evaluation in WCGB benchmark

**System:** You are an advanced AI model equipped with OCR and image understanding capabilities, capable of analyzing visual elements in detail.

**User:** Your task is to assess two webpage images and output a score between 0 and 10 for each of the following questions.

If the answer to a question is a definite YES, output a score of 10, signifying perfect similarity. Conversely, a definite NO should yield a score of 0, indicating no similarity.

For answers that fall in between, assign a score accordingly, where a higher number indicates a greater degree of similarity. Only provide the numerical score for each question, without any additional text. Example contexts are provided for clarity. Examples provides the idea, but you can output any number in 0-10 range accordingly.

Only output a comma separated list containing 10 numbers. DO NOT give score of 10 for any category unless otherwise the two images are identical.

Layout Consistency (Score: 0-10): Does the placement of headers, footers, and sidebars match in both webpages? (e.g., A score of 10 for identical layouts, 5 for similar but not exact placements, and 0 for completely different layouts.)

Element Alignment (Score: 0-10): Are elements like images, buttons, and text boxes aligned similarly on both pages? (e.g., A score of 10 for perfectly aligned elements, 6 for slight misalignments, and 0 for major misalignments.)

Proportional Accuracy (Score: 0-10): Do the sizes and aspect ratios of images, buttons, and text boxes appear consistent across both pages? (e.g., A score of 10 for exact proportions, 4 for noticeable size differences, and 0 for drastic inconsistencies.)

Visual Harmony (Score: 0-10): Do both webpages exhibit a similar level of visual harmony and balance in their design? (e.g., A score of 10 for harmonious designs, 5 for some dissonance, and 0 for clashing designs.)

Color Scheme and Aesthetic Match (Score: 0-10): How closely do the color schemes of the two webpages align in terms of background and text colors? Evaluate the similarity in hues, saturation, and overall color aesthetics. (e.g., A score of 10 for perfectly matching color schemes, including identical hues and saturation levels, 6 for similar color palettes with minor variations, and 0 for starkly different color schemes that create entirely different visual impacts.)

Aesthetic Resemblance (Score: 0-10): Is the overall aesthetic appeal (modern, minimalistic, traditional, etc.) similar on both pages? (e.g., A score of 10 for identical aesthetics, 4 for somewhat similar but distinguishable styles, and 0 for completely different aesthetics.)

Font Characteristics and Consistency (Score: 0-10): Assess the degree of consistency in font attributes across both webpages. This includes not only the font type and size but also the nuances of font style (italic, bold) and weight (light, regular, bold). (e.g., A score of 10 for complete uniformity in font type, size, style, and weight across both pages, 5 for consistency in font type and size but variations in style or weight, and 0 for wide disparities in font type, size, style, or weight, leading to a distinctly different textual appearance.)

Textual Content Match (Score: 0-10): Do the words and sentences match between the two webpages? (e.g., A score of 10 for identical text, 5 for some similar paragraphs or sections, and 0 for completely different textual content.)

Numeric and Special Character Accuracy (Score: 0-10): Are numbers, dates, and special characters (like email addresses) consistent between the two pages? (e.g., A score of 10 for exact matches, 6 for minor discrepancies, and 0 for major differences.)

User Interface Consistency (Score: 0-10): Do the user interface elements (like menus, buttons, and forms) on both pages share a similar design language and appearance? (e.g., A score of 10 for identical UI elements, 6 for slight design variations, and 0 for completely different UI designs.)

<GROUND TRUTH IMAGE>
<PREDICTED IMAGE>

Figure 15: Prompt utilized for evaluation in WCGB with the employment of GPT4-Vision.

### G.4 Prompt Used For QA Generation for WUB benchmark

**User:** Generate 5 QA pairs for assessing an AI model's comprehension and response accuracy in relation to a given webpage screenshot.

The answers should be evenly balanced between YES and NO. These QA pairs will be used as a benchmark to evaluate another model's understanding of the webpage's visual and textual elements.

Ensure the output is formatted as QA pairs in the following dictionary format: '{ 'Q': \"question\", 'A': \"answer\"}'.

The QA pairs should cover the following aspects:

– Advanced Logical Reasoning and Textual Understanding: The questions should challenge the AI's ability to understand complex text and infer meanings or implications that are not explicitly stated.
– Visual Structure and Alignment Evaluation: Questions should test the AI's ability to interpret and analyze the webpage's layout, including the positioning and relationship of various visual elements.
– Color and Aesthetic Design Analysis: Include questions that assess the AI's understanding of the webpage's color scheme and overall design aesthetics, and how they contribute to the page's purpose or message.
– Consistency in Text and Content: Formulate questions that evaluate the AI's recognition of textual consistency and coherence across different sections of the webpage.
– User Interface and Interactivity Insights: Develop questions to assess the AI's comprehension of the webpage's interactive elements and user interface design, including navigational features and response mechanisms.

Remember, the goal is to create questions that will test another AI model's ability to analyze and interpret a webpage screenshot in a comprehensive manner. Do not include additional text other than the output dictionary entry.

Figure 16: Prompt employed to generate "yes" / "no" Question Answer pairs for the WUB benchmark through the utilization of GPT4-Vision.

## H   Data samples

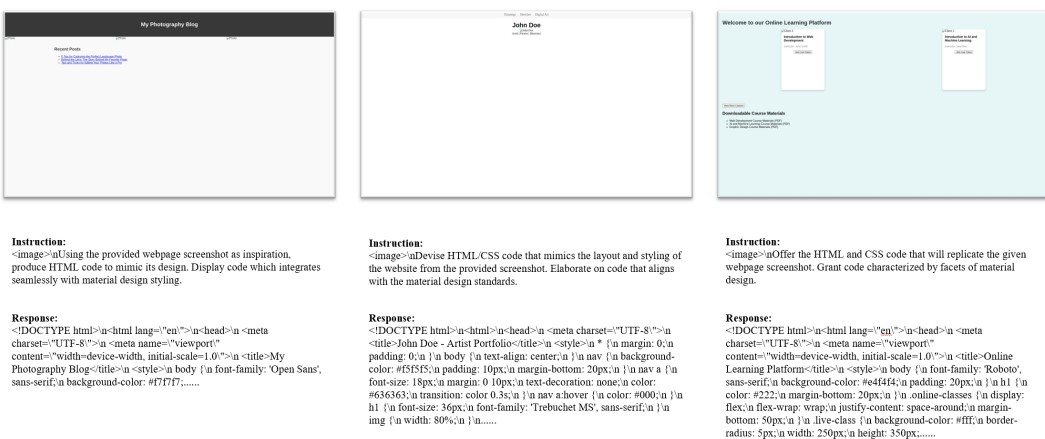

Figure 17: Examples of webpage to code generation instruction tuning data. These web image-code pairs were converted into an instruction-following data format close to the LLaVA data format.

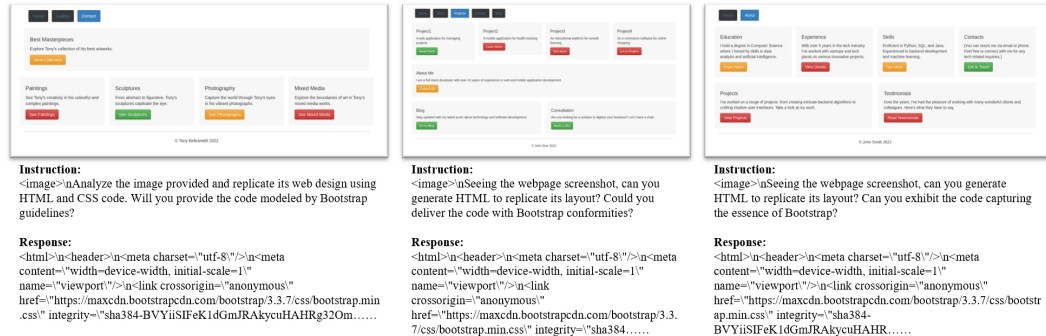

Figure 18: Examples of webpage to code generation instruction tuning data in DWCG. These web image-code pairs were converted into an instruction-following data format close to the LLaVA data format.

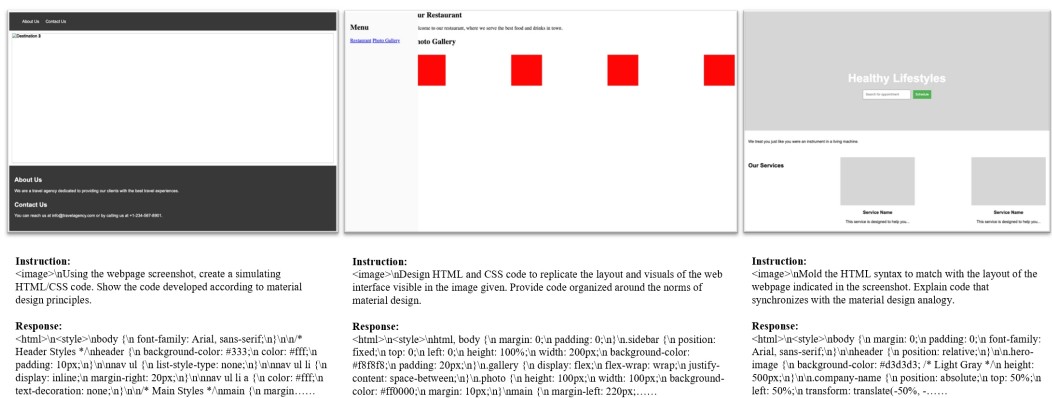

Figure 19: Examples of webpage to code generation instruction tuning data in DWCG$_R$. The refined instruction tuning dataset for webpage code generation, by utilizing GPT-4.

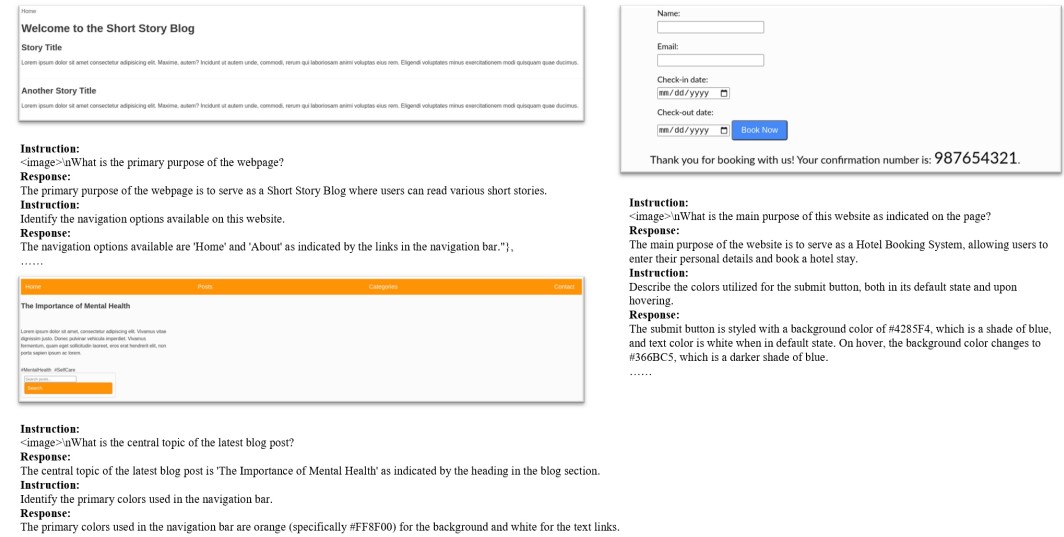

Figure 20: Examples of webpage understanding data in DWU. For a webpage, there is a set of diverse question-answer pairs about web understanding.

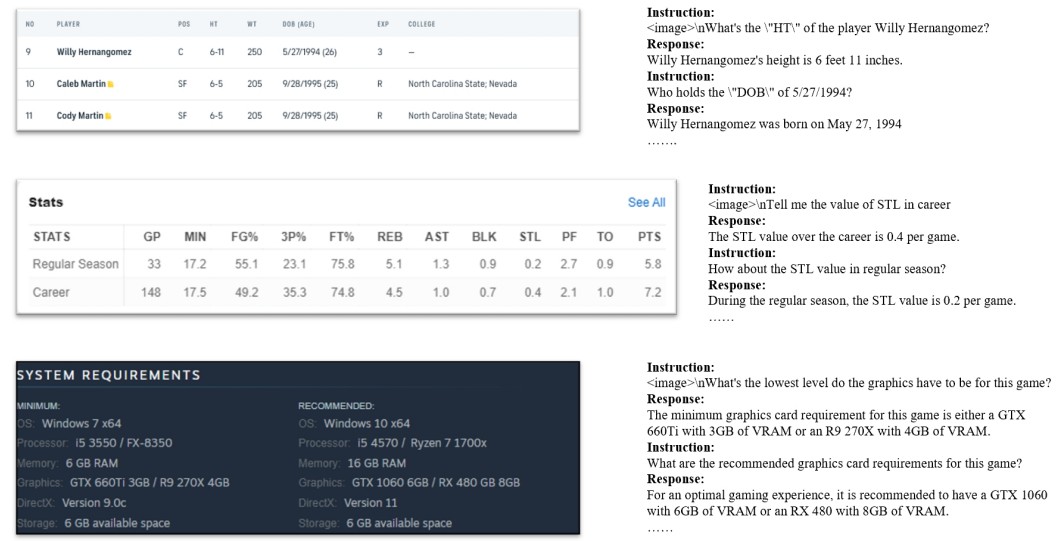

**Instruction:**
<image>\nWhat's the \"HT\" of the player Willy Hernangomez?
**Response:**
Willy Hernangomez's height is 6 feet 11 inches.
**Instruction:**
Who holds the \"DOB\" of 5/27/1994?
**Response:**
Willy Hernangomez was born on May 27, 1994
.......

**Instruction:**
<image>\nTell me the value of STL in career
**Response:**
The STL value over the career is 0.4 per game.
**Instruction:**
How about the STL value in regular season?
**Response:**
During the regular season, the STL value is 0.2 per game.
......

**Instruction:**
<image>\nWhat's the lowest level do the graphics have to be for this game?
**Response:**
The minimum graphics card requirement for this game is either a GTX 660Ti with 3GB of VRAM or an R9 270X with 4GB of VRAM.
**Instruction:**
What are the recommended graphics card requirements for this game?
**Response:**
For an optimal gaming experience, it is recommended to have a GTX 1060 with 6GB of VRAM or an RX 480 with 8GB of VRAM.
......

Figure 21: Examples of webpage understanding data in DWU$_R$. The refined instruction tuning dataset for web understanding tasks.

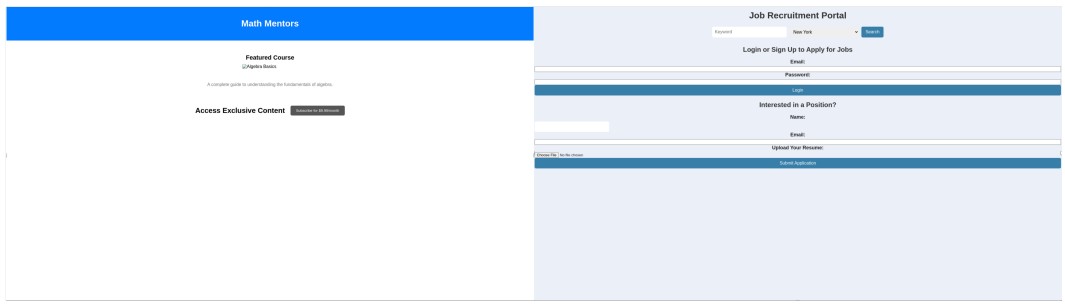

Figure 22: Examples of Modern styles webpages.

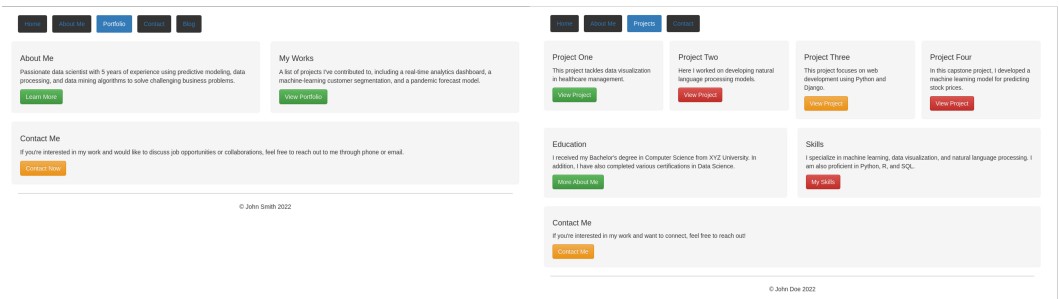

Figure 23: Examples of Bootstrap styles webpages.

