# OpenReview forum: "Web2Code: A Large-scale Webpage-to-Code Dataset and Evaluation Framework for Multimodal LLMs"
_NeurIPS.cc/2024/Datasets_and_Benchmarks_Track — NeurIPS 2024 Track Datasets and Benchmarks Poster_

### Official Review · Reviewer_r3wQ · 2024-07-23

**Rating:** 5
**Confidence:** 5

**Review:**

This paper constructs four meaningful training datasets and two evaluation benchmarks, organizes several experiments, and evaluates different MLLMs under the LLaVa-Style framework, providing some insights. The authors aim to address the current limitations of MLLMs in parsing webpage screenshots and generating corresponding HTML code, and attempt to promote the development of MLLMs in web-based content generation and task automation. However, I have concerns about the quality of the data. Using a stronger model to synthesize data is a common approach in the research community and is also quite straightforward. For example, WebSight0.1 and 0.2 use Deep Seek Coder to synthesize data. The authors do not explain how they control the quality and diversity of the synthesized data, which I think requires additional experiments or methods. Moreover, the authors need to add some experiments to demonstrate the superiority of the synthesized data and some of their claims (please refer to the "Opportunities For Improvement" section for specific content). Overall, the paper has many merits but also exhibits deficiencies in data quality control and lacks some experimental results, necessitating further improvement.

**Strengths:**

1. The Web2Code dataset impresses with its large scale (1.18 million examples) and diversity, focusing on challenging web understanding and HTML generation tasks.
2. The paper introduces two evaluation benchmarks, including web understanding and HTML code generation, providing a more comprehensive evaluation of model performance.
3. Models trained on the Web2Code dataset show improvements in both webpage-to-code translation and general visual domain tasks, demonstrating the dataset's utility in enhancing MLLM capabilities.
4. The dataset and benchmarks help accelerate automated web development, UI prototype design, and contribute to the progress of MLLMs in the coding domain.

**Additional Feedback:**

None.

**Clarity:**

Well-structured and well-written; consider moving some of the main result into the body of the paper. How many pieces of data does the WCGB benchmark contain and where can I find them?

**Correctness:**

The motivation is reasonable, and the dataset construction method is reasonable. However, the data samples are basically synthesized by GPT-3.5/4, and I don’t see how the authors control the data quality.

**Documentation:**

I have reviewed the dataset and benchmark documentation and details, confirming they provide sufficient information to support data collection and organization, availability, maintenance, and ethical and responsible use. Additionally, the benchmark experiments offer enough detail to ensure reproducibility. All submitted content is without issue.

**Ethics:**

None.

**Limitations:**

1. The authors state the limitations of this paper, noting that most datasets are generated using the GPT model. This might introduce biases or limitations in the data, which the paper does not fully discuss.
2. The authors include ethical considerations. Additionally, the potential social impact includes the influence on front-end engineers and the potential misuse of automated webpage generation.

**Opportunities For Improvement:**

1. In the DWU dataset, the authors use GPT-4 to synthesize data. It is unclear whether they input webpage screenshots as shown in Figure 2, or only HTML code and prompts as shown in Figure 11. This could easily be misunderstood as the authors also inputting images. The authors do not clearly describe how they control the quality of synthetic data, which is a concern since simply using GPT-X series models to synthesize data is very straightforward. Additionally, the heavy reliance on synthetic data generated by GPT-3.5 and GPT-4 may limit the dataset's diversity and representativeness.
2. The authors only use a GPT-4V-based evaluation method in the WCGB benchmark, which is too singular and subjective. GPT-4V is continuously updated, and some research finds that models may prefer their own generated results. Consider adding more objective evaluation metrics, such as rule-based metrics mentioned in Design2Code, or CLIP.
3. The baseline settings in Tables 3 and 4 of the evaluation experiments are unusual. There is no report on the performance changes of a single model when incrementally adding training sets. For LLaMA3, only results after training on all four datasets are shown. How does performance change after each dataset is added?
4. The authors repeatedly mention in the paper that "existing datasets lead to performance degradation" without providing any experimental results to support this claim. How would WUB and WCGB change if trained using WebSight0.1 and 0.2 separately? Such baseline results would be more meaningful and better demonstrate the effectiveness of your dataset. The lack of comparison with other recent works on generating code from images or screenshots makes it difficult to evaluate the relative contribution.

**Relation To Prior Work:**

The authors discuss the differences from previous work.

**Summary And Contributions:**

This paper proposes multiple training datasets and evaluation benchmarks to improve the performance of MLLMs in understanding webpage screenshots and generating corresponding HTML code. Web2Code includes over 1.17 million instruction-response pairs, comprising webpage images, HTML code, and NL-Q&A about webpage content, primarily using newly generated data from GPT-3.5/4 and improved existing datasets. The evaluation framework contains two benchmarks: the Web Understanding Benchmark (WUB) with 5,990 yes/no Q&A pairs and the Web Code Generation Benchmark (WCGB) using GPT-4V to evaluate the fidelity of generated webpages. The authors claim that models trained on the Web2Code dataset outperform those trained on previous datasets in both web understanding and HTML code generation tasks.

---

> ### Author Rebuttal · Authors · 2024-08-17
>
> >**Q1: In the DWU dataset, the authors use GPT-4 to synthesize data. It is unclear whether they input webpage screenshots as shown in Figure 2, or only HTML code and prompts as shown in Figure 11. This could easily be misunderstood as the authors also inputting images. The authors do not clearly describe how they control the quality of synthetic data, which is a concern since simply using GPT-X series models to synthesize data is very straightforward. Additionally, the heavy reliance on synthetic data generated by GPT-3.5 and GPT-4 may limit the dataset's diversity and representativeness.**
>
> For clarification, we apologize for the oversight in the details. To generate the DWU question-answer pair data, we used only the HTML code shown in Figure 11 of the original paper, which includes the compiled HTML image. Figure 2 shows the compiled HTML image alongside the corresponding question-answer pairs. For training, we input the image and questions together, excluding the HTML code. This DWU data enhances webpage understanding capabilities.
>
> To ensure both quality and diversity in the synthetic data generated by GPT-3.5 and GPT-4, we employ several strategies. We utilize advanced prompting techniques inspired by CodeAlpaca [3] and incorporate a specialized prompting mechanism based on 10 detailed criteria, as detailed in Section D of the appendix. This approach helps generate a diverse range of data samples.
> Additionally, we integrate multiple existing datasets, such as WebSight [1] and WebSRC [4], to further enrich diversity. Our evaluation process also includes data from Pix2Code [2] and other manually sourced datasets not involved in training, validating our model's generalization capabilities.
> While these measures significantly enhance dataset diversity and quality, we acknowledge a potential gap between the synthetic data and the latest real-world website data, as noted in the limitations section of our original paper.  Table 1 in the response to Reviewer iHjX's Q1 also illustrates the comprehensiveness and diversity of our dataset that covers various categories.
> We will revise our paper carefully to make this clearer.
>
> [1] Hugo Laurençon, Léo Tronchon, and Victor Sanh. Unlocking the conversion of web screenshots into html code with the websight dataset. arXiv preprint arXiv:2403.09029, 2024.
>
> [2] Tony Beltramelli. pix2code: Generating code from a graphical user interface screenshot. arXiv preprint arXiv:1705.07962, 2017.
>
> [3] Sahil Chaudhary. Code alpaca: An instruction-following llama model for code generation. https://github.com/sahil280114/codealpaca, 2023.
>
> [4] Xingyu Chen, Zihan Zhao, Lu Chen, Danyang Zhang, Jiabao Ji, Ao Luo, Yuxuan Xiong, and Kai Yu. Websrc: A dataset for web-based structural reading comprehension. arXiv preprint arXiv:2101.09465, 2021.222

---

> > ### Author Rebuttal · Authors · 2024-08-17
> >
> > >**Q2: The authors only use a GPT-4V-based evaluation method in the WCGB benchmark, which is too singular and subjective. GPT-4V is continuously updated, and some research finds that models may prefer their own generated results. Consider adding more objective evaluation metrics, such as rule-based metrics mentioned in Design2Code, or CLIP.**
> >
> > Thanks for your valuable comments. We understand the concern of this reviewer is the limitations of relying solely on a GPT-4V-based evaluation in the WCGB benchmark. In response to your suggestion, we have incorporated low-level element matching metrics from Design2Code and CLIP-based assessments for visual-text alignment, providing a more comprehensive evaluation. As demonstrated in Table 2, our proposed DWCG method shows significant improvements across various metrics, with the DWU dataset enhancing performance in block-match, position, and color attributes. Models trained with DWCG_R  and DWU_R further demonstrate improvements across all aspects. We aim to integrate these additional metrics into future iterations of the benchmark to address the concerns raised.
> >
> > Table 2. The LLaMA3-8B results on low-level element matching (Design2Code) metrics and CLIP score.
> > | LLM Backbone | DWCG | DWU | DWCG$_{\text{R}}$ | DWU$_{\text{R}}$ | Block-Match $\uparrow$ | Text $\uparrow$ | Position $\uparrow$ | Color $\uparrow$ | CLIP $\uparrow$ |
> >  |------------|:--------:|:------:|:------:|:------:|:------:|------:|------:|------:|------:|
> >  | LLaMA3-8B | - | - | - | - | 5.3 | 11.6 | 7.4 | 26.0 | 73.2 |
> >  | LLaMA3-8B | $\checkmark$ | - | - | - | 76.3 | 92.0 | 74.8 | 63.7 | 84.2 |
> >  | LLaMA3-8B | $\checkmark$ | $\checkmark$ | - | - | 75.4 | 91.3 | **77.3** | 64.9 | 85.6 |
> >  | LLaMA3-8B | $\checkmark$ | $\checkmark$ | $\checkmark$ | - | 77.1 | 94.8 | 76.7 | 64.4 | 86.3 |
> >  | LLaMA3-8B | $\checkmark$ | $\checkmark$ | $\checkmark$ | $\checkmark$ | **77.1** | **96.9** | 77.0 | **66.2** | **86.7** |

---

> > ### Author Rebuttal · Authors · 2024-08-17
> >
> > >**Q3: The baseline settings in Tables 3 and 4 of the evaluation experiments are unusual. There is no report on the performance changes of a single model when incrementally adding training sets. For LLaMA3, only results after training on all four datasets are shown. How does performance change after each dataset is added?**
> >
> > We appreciate your valuable comments. Our initial focus was on demonstrating the effectiveness of the proposed training sets on various ablation scenarios among various LLM backbones. For example, CrystalCoder results show the effects of incrementally adding training sets by DWCG, DWU, and refined ones (i.e., DWCG_R and DWU_R).
> >
> > To make our ablation study more comprehensive, we have incorporated additional results on LLaMA3 as suggested. As shown in Table 3, we observe that the proposed web2code dataset significantly enhances overall performance across both WUB and WCGB. For example, the DWCG dataset improves code generation capabilities, though it requires more webpage understanding. However, the DWU dataset not only recovers but also enhances both WUB and WCGB performances. Similarly, the refined dataset DWCG_R primarily boosts WCGB, while DWU_R shows improvements across all metrics. We will include these more results and findings in our revision.
> >
> > Table 3. The LLaMA3-8B results on WCGB and WUB.
> > | LLM Backbone | DWCG | DWU | DWCG$_{\text{R}}$ | DWU$_{\text{R}}$ | VSA $\uparrow$ | CAD $\uparrow$ | TCC $\uparrow$ | UII $\uparrow$ | WCGB Overall $\uparrow$ | WUB Accuracy |
> >  |------------|:--------:|:------:|:------:|:------:|:------:|------:|------:|------:|------:|------:|
> >  | LLaMA3-8B | - | - | - | - | 1.563 | 1.777 | 1.894 | 1.911 | 1.79 | 65.56 |
> >  | LLaMA3-8B | $\checkmark$ | - | - | - | 5.613 | 6.575 | 6.551 | 6.870 | 6.402 | 60.00 |
> >  | LLaMA3-8B | $\checkmark$ | $\checkmark$ | - | - |6.564 | 6.762 | 6.998 | 6.541 | 6.716 | 69.33 |
> >  | LLaMA3-8B | $\checkmark$ | $\checkmark$ | $\checkmark$ | - | 7.667 | 7.560 | 7.995 | 8.001 | 7.806 | 68.68 |
> >  | LLaMA3-8B | $\checkmark$ | $\checkmark$ | $\checkmark$ | $\checkmark$ | **8.522** | **8.564** | **8.421** | **8.611** | **8.530** | **74.84** |
> >
> > >**Q4: The authors repeatedly mention in the paper that "existing datasets lead to performance degradation" without providing any experimental results to support this claim. How would WUB and WCGB change if trained using WebSight0.1 and 0.2 separately? Such baseline results would be more meaningful and better demonstrate the effectiveness of your dataset. The lack of comparison with other recent works on generating code from images or screenshots makes it difficult to evaluate the relative contribution.**
> >
> > Thanks for the suggestions. We would like to clarify that our main goal is to enhance not only webpage (i.e., HTML code) generation capability from webpage screenshots (i.e., WCGB), but also webpage screenshot understanding (i.e., WUB) as they are closely related. To this end, existing single HTML code generation datasets like WebSight are insufficient to achieve both simultaneously. Moreover, the existing public datasets would require additional refinements for privacy protection or being more realistic. For example, as we respond to Reviewer iHjX's Q2, WebSight dataset would contain privacy-sensitive information such as phone numbers, email addresses, credit card numbers, and personal images. Also, the original Pix2Code dataset contains random letters in the whole web screenshots, which harms the ability of MLLMs to generate reasonable texts for webpage generation.
> >
> > Following the reviewer's suggestions, we have provided more comparisons with additional baselines of the WebSight datasets. In Table 4, as we have pointed out above, the single HTML code generation dataset (i.e., WebSight0.1) significantly drops the performances of WUB compared to the original LLaVA dataset (without any additional code-related data). We also observe that even a single DWCG dataset (our 60K generated webpage code generation data) outperforms the Websight0.1 dataset of 823K. Table 5 also shows the effectiveness of the proposed dataset DWCG and DWU. Similar to the trend shown in Table 4, our 60K generated data, DWCG, significantly outperforms ~14 times larger dataset baseline, WebSight0.1, across all low-level element matching metrics (Design2Code) and CLIP score.
> >
> > Furthermore, Tables 6, 7, and 8 summarize comparisons between the Webstight dataset and the proposed dataset, Web2Code, on various evaluation metrics, including WCGB, WUB, Design2Code, CLIP, and general domain benchmarks. Overall, our Web2Code dataset significantly outperforms the WebSight baselines across all metrics. We would like to clarify that such degradation of existing code generation datasets in the general domain motivates us to explore ways to refine them and simultaneously collect (or generate) both code generation data and webpage understanding data to maintain overall performance. We will include those results in our revised paper.
> >
> > Table 4. The WebSight baseline results on WCGB and WUB.
> >
> >  | LLM Backbone | DWCG | DWU | WebSight0.1 | WebSight0.2 | VSA $\uparrow$ | CAD $\uparrow$ | TCC $\uparrow$ | UII $\uparrow$ | WCGB Overall $\uparrow$ | WUB Accuracy |
> >  |------|:-----:|:------:|:------:|:------:|:------:|------:|------:|------:|------:|------:|
> >  | CrystalCoder-7B | - | - | - | - | 3.832 | 3.678 | 3.411 | 3.992 | 3.728 | 73.54 |
> >  | CrystalCoder-7B | $\checkmark$ | - | - | - | 7.812 | 7.899 | 8.138 | 8.112 | 7.990 | 71.81 |
> >  | CrystalCoder-7B | $\checkmark$ | $\checkmark$ | - | - | **8.010** | **8.102** | **8.266** | **8.124** | **8.126** | **73.74** |
> >  | CrystalCoder-7B | - | - | $\checkmark$ | - | 7.524 | 7.600 | 7.818 |	7.862 | 7.701 | 59.38 |
> >  | CrystalCoder-7B | - | - | - | $\checkmark$ | 5.469 | 5.830 | 6.113 | 5.911 | 5.831 | 56.06 |

---

> > ### Author Rebuttal · Authors · 2024-08-17
> >
> > Table 5. The WebSight baseline results on low-level element matching (Desgin2Code) metrics and CLIP score.
> >
> >  | LLM Backbone | DWCG | DWU | WebSight0.1 | WebSight0.2 | Block-Match $\uparrow$ | Text $\uparrow$ | Position $\uparrow$ | Color $\uparrow$ | CLIP $\uparrow$ |
> >  |------|:------:|:------:|:------:|:------:|:------:|:------:|:------:|:------:|:------:|
> >  | CrystalCoder-7B | - | - | - | - | 8.6  | 13.2 | 6.5 | 17.4 | 70.3 |
> >  | CrystalCoder-7B | $\checkmark$ | - | - | - | **76.6** | 90.4 | **78.2** | 63.1 | **85.2** |
> >  | CrystalCoder-7B | $\checkmark$ | $\checkmark$ | - | - | 75.4 | **91.7** | 77.4 | **64.0** | 83.9 |
> >  | CrystalCoder-7B | - | - | $\checkmark$ | - | 62.4 | 77.7 | 63.9 | 60.7 | 79.2 |
> >  | CrystalCoder-7B | - | - | - | $\checkmark$ | 61.8 | 78.2 | 62.9 | 62.1 | 78.7 |
> >
> > Table 6. Comparisons between the WebSight datasets and the proposed Web2Code dataset on WCGB and WUB.
> >
> >  | LLM Backbone | Web2Code (Ours) | WebSight0.1 | WebSight0.2 | VSA $\uparrow$ | CAD $\uparrow$ | TCC $\uparrow$ | UII $\uparrow$ | WCGB Overall $\uparrow$ | WUB Accuracy |
> >  |------------|:--------:|:------:|:------:|:------:|:------:|:------:|:------:|:------:|:------:|
> >  | LLaMA3-8B | - | - | - | 1.563 | 1.777 | 1.894 | 1.911 | 1.790 | 65.56 |
> >  | LLaMA3-8B | $\checkmark$ | - | - | **8.522** | **8.564** | **8.421** | **8.611** | **8.530** | **74.84** |
> >  | LLaMA3-8B | - | $\checkmark$ | - | 4.236 | 4.113 | 3.981 | 4.168 | 4.125 | 70.13 |
> >  | LLaMA3-8B | - | - | $\checkmark$ | 4.611 | 5.238 | 5.923 | 4.689 | 5.115 | 57.33 |
> >
> > Table 7. Comparisons between the WebSight datasets and the proposed Web2Code dataset on low-level element matching (Desgin2Code) metrics and CLIP score.
> >
> >  | LLM Backbone | Web2Code (Ours) | WebSight0.1 | WebSight0.2 | Block-Match $\uparrow$ | Text $\uparrow$ | Position $\uparrow$ | Color $\uparrow$ | CLIP $\uparrow$ |
> >  |------------|:--------:|:------:|:------:|:------:|:------:|:------:|:------:|:------:|
> >  | LLaMA3-8B | - | - | - | 5.3 | 11.6 | 7.4 | 26.0 | 73.2 |
> >  | LLaMA3-8B | $\checkmark$ | - | - |  **77.1** | **96.9** | **77.0** | **66.2** | **86.7** |
> >  | LLaMA3-8B | - | $\checkmark$ | - | 60.1 | 78.5 | 60.9 | 60.3 | 79.5 |
> >  | LLaMA3-8B | - | - | $\checkmark$ | 57.3 | 73.9 | 59.1 | 60.8 | 76.1 |
> >
> >  Table 8. Comparisons between the WebSight datasets and the proposed Web2Code dataset on general domain benchmarks.
> >
> >  | LLM Backbone | Web2Code (Ours) | WebSight0.1 | WebSight0.2 | MME-P | MME-C | POPE | SciQA | TextVQA |
> >  |------------|:--------:|:------:|:------:|:------:|:------:|:------:|:------:|:------:|
> >  | LLaMA3-8B | $\checkmark$ | - | - | **1464.73** | 265.14 | **86.34** | **72.08** | **58.76** |
> >  | LLaMA3-8B | - | $\checkmark$ | - | 1349.33 | 235.47 | 85.73 | 61.42 | 23.38 |
> >  | LLaMA3-8B | - | - | $\checkmark$ | 1149.68 | **270.36** | 83.29 | 56.19 | 20.51 |

---

> > ### Author Rebuttal · Authors · 2024-08-17
> >
> > >**L1: The authors state the limitations of this paper, noting that most datasets are generated using the GPT model. This might introduce biases or limitations in the data, which the paper does not fully discuss.**
> >
> > Thank you for raising this concern. GPT models, like any LLMs, can reflect biases present in their training data. This means that the synthetic data generated may inherit some of these biases, potentially leading to skewed representations in the dataset. To mitigate this, we have employed advanced prompting techniques as described in the paper to guide GPT's generation process towards a more balanced output. Additionally, we have incorporated data from multiple sourced datasets to counterbalance any potential bias from the synthetic data. We will include these discussions in our revision.
> >
> > >**L2: The authors include ethical considerations. Additionally, the potential social impact includes the influence on front-end engineers and the potential misuse of automated webpage generation.**
> >
> > Thanks for the acknowledgment and for pointing out more potential social impact. The automation of webpage generation through advanced models like ours may lead to concerns about the displacement of front-end engineering jobs. However, we view this technology as a tool to augment, rather than replace, the work of engineers. By automating repetitive tasks, front-end engineers can focus more on creative and complex aspects of web development, leading to higher productivity and innovation. We also recognize that automated webpage generation could be misused, for example, in creating deceptive or harmful websites. To mitigate this risk, we advocate for the development of clear ethical guidelines and best practices for the use of such technology. We encourage users and organizations to deploy our models responsibly, ensuring that the generated content is accurate, transparent, and not intended to deceive or harm users. Moreover, we are committed to ongoing research into methods for detecting and preventing misuse, including incorporating safeguards into our models. We will include these discussions in our revised paper.
> >
> > >**C: The motivation is reasonable, and the dataset construction method is reasonable. However, the data samples are basically synthesized by GPT-3.5/4, and I don’t see how the authors control the data quality.**
> >
> > To ensure both quality and diversity in the synthetic data generated by GPT-3.5 and GPT-4, we employ several strategies. We utilize advanced prompting techniques inspired by CodeAlpaca [3] and incorporate a specialized prompting mechanism based on 10 detailed criteria, as detailed in Section D of the appendix. This approach helps generate a diverse range of data samples.
> > Additionally, we integrate multiple existing datasets, such as WebSight [1] and WebSRC [4], to further enrich diversity. Our evaluation process also includes data from Pix2Code [2] and other manually sourced datasets not involved in training, validating our model's generalization capabilities.
> >
> > While these measures significantly enhance dataset diversity and quality, we acknowledge a potential gap between the synthetic data and the latest real-world website data, as noted in the limitations section of our original paper.  Table 1 in the response to Reviewer iHjX's Q1 also illustrates the comprehensiveness and diversity of our dataset that covers various categories.
> > We will revise our paper carefully to make this clearer.
> >
> > [1] Hugo Laurençon, Léo Tronchon, and Victor Sanh. Unlocking the conversion of web screenshots into html code with the websight dataset. arXiv preprint arXiv:2403.09029, 2024.
> >
> > [2] Tony Beltramelli. pix2code: Generating code from a graphical user interface screenshot. arXiv preprint arXiv:1705.07962, 2017.
> >
> > [3] Sahil Chaudhary. Code alpaca: An instruction-following llama model for code generation. https://github.com/sahil280114/codealpaca, 2023.
> >
> > [4] Xingyu Chen, Zihan Zhao, Lu Chen, Danyang Zhang, Jiabao Ji, Ao Luo, Yuxuan Xiong, and Kai Yu. Websrc: A dataset for web-based structural reading comprehension. arXiv preprint arXiv:2101.09465, 2021.222

---

> > ### Author Rebuttal · Authors · 2024-08-28
> >
> > Dear Reviewer r3wQ,
> >
> > &nbsp;
> >
> > Thanks for your patience. We have provided all the requested additional results, baseline experiments, and clarifications. Please check them out and let us know if you have any further questions or suggestions. We sincerely appreciate your thorough review, constructive feedback, and the time and effort you invested in reviewing our paper. We will accommodate all of your comments in our revised paper.
> >
> > &nbsp;
> >
> > Best,
> >
> > Authors

---

### Official Review · Reviewer_iHjX · 2024-07-24
**Review by Reviewer iHjX**

**Rating:** 7
**Confidence:** 3
**Correctness:** The dataset is constructed relatively…
**Clarity:** The writing of the paper is relativel…

**Review:**

See [Summary].

**Strengths:**

1. This paper proposes a new large-scale web-to-code dataset and evaluation framework, providing new challenges and evaluation criteria for multi-modal LLMs in web page understanding and HTML code generation.

2. The author used pre-trained LLMs to construct a data set containing diverse natural language question and answer pairs, thereby obtaining a higher quality and richer data set, which helps to improve the model's performance on web page understanding and code generation tasks.

3. The authors propose methods to effectively evaluate the model's capabilities in web page understanding and HTML code parsing, while also evaluating the quality of the generated code.

**Additional Feedback:**

None.

**Documentation:**

There is relatively detailed documentation.

**Ethics:**

The authors require further detailed statements regarding their protection of privacy issues.

**Limitations:**

See [Opportunities For Improvement].

**Opportunities For Improvement:**

1. Although the dataset is large, it may still be biased and may not cover all possible real-life scenarios.

2. Datasets may involve privacy-sensitive issues. Privacy protection measures during data collection and processing need to be introduced in more detail to avoid potential legal and ethical risks.

**Relation To Prior Work:**

The paper discusses the relationship with previous papers.

**Summary And Contributions:**

This paper proposes the Web2Code dataset and evaluation framework, aiming to improve multi-modal LLMs' understanding of web page screenshots and HTML code generation capabilities. By augmenting existing datasets with pre-trained LLMs and generating new web page images, the authors constructed a dataset containing diverse natural language question-answer pairs.

---

> ### Author Rebuttal · Authors · 2024-08-17
>
> >**Q2: Datasets may involve privacy-sensitive issues. Privacy protection measures during data collection and processing need to be introduced in more detail to avoid potential legal and ethical risks.**
>
> Thank you for raising the critical point of privacy protection in our dataset involving the webpage-to-code task. We fully acknowledge the importance of safeguarding privacy and have implemented several measures to ensure that our data collection and processing adhere to legal and ethical standards. Below, we provide a detailed overview of these privacy protection measures: 1) Personal data removal: During the data collection process, any personal or sensitive information (e.g., names, email addresses, phone numbers) that could be linked to individuals was systematically removed. This anonymization process ensures that the dataset does not contain any information that could directly identify users. 2) Scrubbing of sensitive fields: Fields known to potentially hold sensitive information, such as form inputs or user comments, were either excluded from the dataset or anonymized to prevent privacy breaches. 3) Publicly available data: The dataset was curated and refined primarily from publicly available datasets where no authentication is required. This reduces the risk of capturing sensitive, private content that is not intended for public consumption. 4) Exclusion of personally identifiable information: No personally identifiable information was included in the dataset. Any data fields that could inadvertently contain personally identifiable information were carefully reviewed and excluded or anonymized. To further assess privacy concerns, we conducted a thorough examination during the rebuttal by randomly sampling 1,000 data points from our dataset and performing a human evaluation to check for privacy-sensitive information, such as phone numbers, email addresses, credit card numbers, and personal images. We find that only 2.7% (16 email addresses, 11 phone numbers, and no credit card numbers) of the data contained such issues, indicating the effectiveness of our privacy protection measures.
> We will include these details in our revised paper.

---

> ### Author Rebuttal · Authors · 2024-08-17
>
> >**Q1: Although the dataset is large, it may still be biased and may not cover all possible real-life scenarios.**
>
> Thank you for highlighting this important consideration. While we appreciate the acknowledgment that our webpage-to-code dataset is large, we understand that no dataset can comprehensively cover all real-life scenarios. To address potential biases and gaps, we have implemented several measures: 1) Diversity in data collection: The dataset was curated from a wide variety of sources, including different types of websites and design patterns, to ensure broad coverage of real-life scenarios, Figures 4 and 5 in our submission also illustrate the diversity of our data. 2) Extensive evaluation: Beyond our primary dataset, we have tested our model on external and unseen datasets to evaluate its generalizability and its ability to manage scenarios not directly represented in the training data.
>
> Additionally, we have calculated the distribution percentages of different types of web pages in our dataset, as shown in Table 1 and Figure (in the pdf), to further illustrate its comprehensiveness.
>
> Table 1. Data distribution percentages (%) of different types by category.
> | Corporate | Portfolio | Contact | Personal | Advertisement | Landing | Blog | E-commerce | Educational |
>  |:--------:|:--------:|:------:|:------:|:------:|:------:|:------:|:------:|:------:|
>  | 10.83 | 4.77 | 9.17 | 0.52 | 20.68 | 10.23 | 8.21 | 10.77 | 2.78 |
>
> | Non-profit | Login | Entertainment | News | Forum | Event | Documentation | Marketing | Unknown |
>  |:--------:|:--------:|:------:|:------:|:------:|:------:|:------:|:------:|------:|
>  | 2.54 | 1.02 | 1.74 | 0.69 | 1.89 | 0.81 | 0.15 | 0.05 | 13.14 |

---

### Author Rebuttal · Authors · 2024-08-17

We appreciate all reviewers for their positive comments, e.g., this paper proposes a new large-scale web-to-code dataset and evaluation framework, providing new challenges and evaluation criteria for multi-modal LLMs in web page understanding and HTML code generation [iHjX], the author used pre-trained LLMs to construct a data set containing diverse natural language question and answer pairs, thereby obtaining a higher quality and richer data set, which helps to improve the model's performance on web page understanding and code generation tasks [iHjX], the authors propose methods to effectively evaluate the model's capabilities in web page understanding and HTML code parsing, while also evaluating the quality of the generated code [iHjX], the Web2Code dataset impresses with its large scale (1.18 million examples) and diversity, focusing on challenging web understanding and HTML generation tasks [r3wQ], the paper introduces two evaluation benchmarks, including web understanding and HTML code generation, providing a more comprehensive evaluation of model performance [r3wQ], models trained on the Web2Code dataset show improvements in both webpage-to-code translation and general visual domain tasks, demonstrating the dataset's utility in enhancing MLLM capabilities [r3wQ], the dataset and benchmarks help accelerate automated web development, UI prototype design, and contribute to the progress of MLLMs in the coding domain [r3wQ], and the constructive suggestions, e.g., bias concern and may not cover all possible real-life scenarios [iHjX],  privacy protection measures and details during data collection [iHjX], details whether inputing webpage screenshots [r3wQ], dataset's diversity and representativeness [r3wQ], baseline settings [r3wQ], etc., which will definitely help us improve the quality of this paper.

We will accommodate all of the comments in our revision. In the following, we summarize our responses and clarify further questions from each reviewer.

1. We have provided the details of our implemented measures to mitigate the potential biases and gaps in real-life scenarios, as well as the statistics of data distribution percentages for different types of categories to demonstrate dataset diversity. [iHjX]

2. We have provided a detailed overview of our privacy protection measures, including: Personal data removal; Scrubbing of sensitive fields; Using publicly available data; Exclusion of personally identifiable information. [iHjX]

3. We have clarified the usage of webpage screenshots in the DWU dataset and provided details on how to ensure both quality and diversity in the synthetic data generated by GPT-3.5 and GPT-4 in our framework. [r3wQ]

4. We have incorporated low-level element matching metrics from Design2Code and CLIP-based assessments for visual-text alignment, providing a more comprehensive evaluation. One missing result will be updated in a few days. [r3wQ]

5. We have added ablation results to demonstrate the impact of incrementally adding training sets on LLaMA3. [r3wQ]

6. We have included additional baselines using WebSight to further validate the effectiveness of the proposed dataset. [r3wQ]

7. We have expanded our discussion on potential biases and limitations within the data. [r3wQ]

8. We have also provided more insights into the potential social impact, including the effects on front-end engineers and the risks associated with the misuse of automated webpage generation. [r3wQ]

Please feel free to let us know if the reviewers have any further questions, and we are happy to address them. Thank you for the time and effort you put into the review!

---

### Decision · Program_Chairs · 2024-09-26

**Decision:**

Accept (Poster)

**Comment:**

In the manuscript, the authors propose the Web2Code dataset and use it to benchmark performance of models in web understanding and HTML code generation.

First of all, it was highly unfortunate that only 2 reviews were received and both reviewers did not participate in the post-review discussions. This meta review attempts to take these factors into account but is fundamentally limited by the lack of information provided the reviewers.

Strengths:
1. The dataset constructed is large and contains diverse types of web pages.
2. The two benchmarks performed provide a good demonstration of the dataset.
3. Some models showed performance improvements when trained on the Web2Code dataset.

Weaknesses:
1. A reviewer questioned about the quality and diversity of the synthesized data. The authors provided clarifications during the rebuttal, which were unfortunately not further commented on by the reviewer. Yet the authors also admit that synthesized data have limitations and have added discussion of it to the manuscript.
2. A reviewer commented on the use of only GPT-4V to evaluation results in the WCGB benchmark. The authors added extra results in response. Whether they were sufficient for addressing the comment was, once again, unfortunately not confirmed by the reviewer.
3. There were comments on claims made in the manuscript the supporting evidence of which was not clearly pointed out.

Based on the detailed rebuttals provided by the authors, most concerns raised by the reviewers appear to have been addressed. Therefore, the Area Chair believes at least one of the reviewers would have increased the score by at least one point if they participated in the post-review discussions, which would have made the overall score reaching the standard for acceptance.